# Overthinking Reduction with Decoupled Rewards and Curriculum Data Scheduling

**Shuyang Jiang**[♠,♣]**, Yusheng Liao**[♢]**, Ya Zhang**[♢,♣,*]**, Yanfeng Wang**[♢,♣]**, Yu Wang**[♢,♣,*]

[♠]Fudan University

[♢]School of Artificial Intelligence, Shanghai Jiao Tong University

[♣]Shanghai Artificial Intelligence Laboratory

shuyangjiang23@m.fudan.edu.cn

{liao20160907,ya_zhang,wangyanfeng622,yuwangsjtu}@sjtu.edu.cn

## Abstract

While large reasoning models trained with critic-free reinforcement learning and verifiable rewards (RLVR) represent the state-of-the-art, their practical utility is hampered by "overthinking", a critical issue where models generate excessively long reasoning paths without any performance benefit. Existing solutions that penalize length often fail, inducing performance degradation due to a fundamental misalignment between trajectory-level rewards and token-level optimization. In this work, we introduce a novel framework, DECS, built on our theoretical discovery of two previously unaddressed flaws in current length rewards: (1) the erroneous penalization of essential exploratory tokens and (2) the inadvertent rewarding of partial redundancy. Our framework's innovations include (i) a first-of-its-kind decoupled token-level reward mechanism that surgically distinguishes and penalizes redundant tokens, and (ii) a novel curriculum batch scheduling strategy to master the efficiency-efficacy equilibrium. Experimental results show DECS can achieve a dramatic reduction in reasoning tokens by over 50% across seven benchmarks while simultaneously maintaining or even improving performance. It demonstrates conclusively that substantial gains in reasoning efficiency can be achieved without compromising a model's underlying reasoning power. Code is available at https://github.com/pixas/DECS.

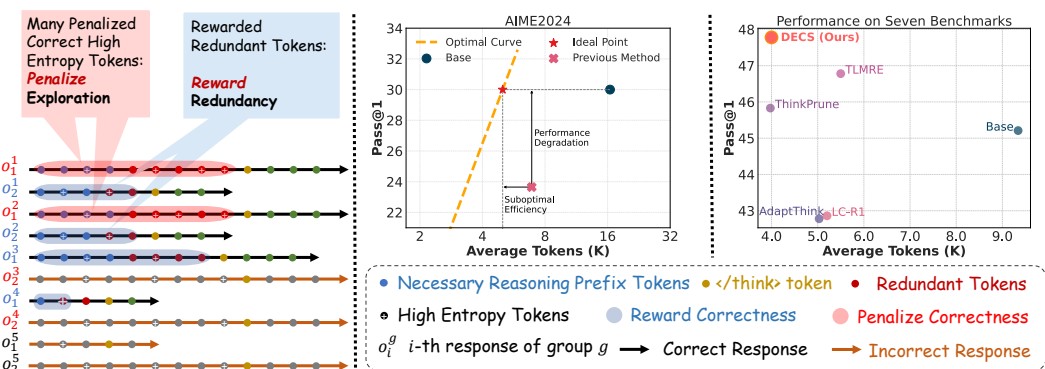

Figure 1: **Left**: Two major flaws of prior practice apply sequence-level length reward without control of training data. Negative advantage values penalize correct high entropy tokens from long sequences while positive ones reward redundant tokens from short sequences; **Middle**: Flaws of length rewards lead to inferior performance and suboptimal efficiency gains on AIME2024 dataset; **Right**: DECS improves pass@1 of base models while reducing $\sim 60\%$ token costs compared to the base model across 7 benchmarks. Experimental details are presented in Appendix G.5.

---

*Corresponding Author

# 1 INTRODUCTION

Recent large reasoning models (LRM; Guo et al. (2025); OpenAI (2025); Qwen (2025)) trained with critic-free reinforcement learning (RL) algorithms, such as GRPO (Shao et al., 2024), DAPO (Yu et al., 2025), and REINFORCE++ (Hu et al., 2025a), have demonstrated impressive reasoning capabilities through verifiable outcome rewards. A hallmark of such models is their increased propensity to generate high-entropy tokens (e.g., "wait", "however", "alternatively"), which serve to bridge logical transitions between reasoning steps (Wang et al., 2025b). While these tokens reflect active reasoning mechanisms that enhance performance, the propagation of trajectory-level rewards to all tokens can inadvertently encourage prolonged generation led by high-entropy tokens even after reaching a correct answer, a phenomenon known as "overthinking" (Ji et al., 2025). To address this inefficiency without sacrificing reasoning quality, recent approaches incorporate a small length penalty into the correctness reward (Hou et al., 2025; Su & Cardie, 2025; Aggarwal & Welleck, 2025; Zhang et al., 2025d; Kimi et al., 2025; Wu et al., 2025), using critic-free RL frameworks like GRPO to promote concise yet effective reasoning.

Despite these advancements, we find that existing methods still fall short of achieving the optimal efficiency-performance trade-off: improvements in reasoning speed often come at the expense of degraded reasoning fidelity. This suboptimality raises a fundamental question: *why do current reward designs fail to effectively balance brevity and capability*? To investigate this, we conduct a theoretical analysis of the logit dynamics of two key groups of tokens within the GRPO framework: (i) high-entropy tokens that initiate exploratory reasoning paths, and (ii) those belonging to the Necessary Reasoning Prefix (NRP), defined as the minimal prefix of a reasoning trajectory that suffices to justify the final correct answer. Our analysis reveals two critical limitations arising from the misalignment between sequence-level length regularization and token-level policy updates (depicted in Fig. 1(Left)), revealing inherent tensions in how efficiency is incentivized during training.

First, sequence-level length penalties inherently suppress high-entropy tokens, even when they contribute to valid reasoning (§3.2). Specifically, in GRPO, overlong (yet correct) trajectories receive uniformly negative advantages across all tokens from length penalties. Consequently, when all responses to a given prompt are correct but differ in length, shorter trajectories yield positive advantages while longer ones receive negative ones. This leads to a reduction in the logits of high-entropy tokens through policy gradient updates. When easy prompts dominate the batch and response lengths vary significantly, this negative gradient becomes dominant over iterations, causing the policy to avoid generating these tokens, even if they are essential for productive exploration (Theorem 1). This leads to premature convergence and deviation from the optimal efficiency-efficacy trade-off.

Second, training convergence is impeded by misaligned incentives (§3.3). Without explicitly decoupling the NRP serving as the minimal sufficient reasoning prefix from subsequent generations, tokens produced after the NRP in shorter trajectories may still receive positive advantages. This falsely reinforces redundant steps, encouraging the model to continue generating beyond logical necessity. These spurious rewards not only distort the learning signal but also slow down convergence, limiting the extent of achievable efficiency gains under finite training updates.

Building on these insights, we propose DECS, a novel framework with **De**coupled token-level rewards and **C**urriculum data **S**cheduling for overthinking reduction (§4). To enable precise intervention, we fine-tune a lightweight judge model to identify NRP boundaries. Based on this, we design a decoupled reward function that ensures redundant tokens generated after the NRP are consistently penalized, thereby suppressing overthinking during autoregressive decoding. Meanwhile, we introduce a curriculum batching strategy that adaptively balances the proportion of easy prompts according to the average NRP ratio in the current batch, mitigating undue suppression of exploratory behavior. Experimental results on two base models show that DECS reduces reasoning tokens by over 50%, while maintaining or surpassing performance on both deterministic (Pass@1; Table 1) and exploratory (Pass@K; Fig. 3c) metrics. In summary, we conclude our contributions as follows:

1. **Misalignment Analysis**: We identify a fundamental misalignment between sequence-level length penalties and token-level policy optimization in critic-free RL. Our theoretical analysis demonstrates that this misalignment not only inhibits the generation of high-entropy tokens, which are essential for valid reasoning, but also hampers efficiency improvements due to misguided gradient signals.

2. **Adaptive Sampling with Decoupled Reward**: We introduce DECS, a novel method that employs a decoupled reward system to consistently penalize redundancy. Coupled with a dynamic batching strategy, this approach mitigates the over-penalization of exploration by incorporating adaptive curriculum control.

3. **Comprehensive Evaluation**: We perform extensive evaluations across two model scales and seven benchmarks, showing that DECS consistently reduces over 50% thinking tokens without sacrificing base models' capability boundary.

## 2 PRELIMINARY

### 2.1 REINFORCEMENT LEARNING WITH VERIFIABLE REWARDS (RLVR)

The RL objective for the policy $\pi_\theta$ is to maximize the cumulative rewards $r$ received from the verifier. Specifically, Policy Gradient (Williams, 1992) gives the following objective function:

$$\nabla \mathcal{J}(\theta) = \mathbb{E}_{q \sim \mathcal{D}, o \sim \pi_\theta(q)} \sum_{j=0}^{T} \nabla_\theta \log \pi_\theta(o_j \mid o_{<j}) A(o_{<j}, j), \tag{1}$$

where $\mathcal{D}$ is the training distribution, $q$ is an input prompt, $o$ is an output sequence consisting of $T$ tokens $\{o_1, o_2, \ldots, o_T\}$, and $A(o_{<j}, j)$ is the advantage of the $j$-th token given the state $o_{<j}$. Recently, DeepSeek-R1 (Guo et al., 2025) boosted large language models' reasoning ability via the Group Relative Policy Optimization (GRPO; Shao et al. (2024)) algorithm. Each rollout is labeled with a verifiable reward $r(\cdot)$, while its advantage is estimated using the group average and standard deviation values of rewards from a group of $G$ trajectories $\mathcal{O} = \{o_i\}_{i=1}^{G}$ generated based on the same prompt $q$:

$$A_i = \frac{r(o_i) - \text{mean}(r(o_1), \ldots, r(o_G))}{\text{std}(r(o_1), \ldots, r(o_G))}. \tag{2}$$

GRPO optimizes the policy using the PPO surrogate loss (Schulman et al., 2017):

$$\mathbb{E}_{q \sim \mathcal{D}, \{o_i\}_{i=1}^{G} \sim \pi_\theta(\cdot|q)} \left[ \frac{1}{\sum_i^G |o_i|} \sum_{i=1}^{G} \sum_{j=1}^{|o_i|} \min\left(\rho_{i,j} A_i, \text{clip}(\rho_{i,j} A_i, 1-\epsilon, 1+\epsilon) A_i\right) \right], \tag{3}$$

where $\rho_{i,j} = \pi_\theta(o_{i,j} \mid o_{i,<j}, q) / \pi_{\text{old}}(o_{i,j} \mid o_{i,<j}, q)$ is the importance sampling ratio, $|o_i|$ is the sequence length. The KL term is reduced to align with Hu et al. (2025b). Models are incentivized to explore new trials, cross-verifying temporary results using diverse approaches, and correct existing results, based on high-entropy decisive tokens (Wang et al., 2025b). However, although the high frequency of generating high-entropy triggers does boost the model for challenging problems (Muennighoff et al., 2025), such improvements are not consistent (Ghosal et al., 2025), and introduce great verbosity and "over-thinking" for vanilla queries (Ji et al., 2025).

### 2.2 EFFICIENT REASONING WITH LENGTH PENALTIES

One of the most straightforward methods is to add a length-based reward along with the fundamental correctness reward to encourage shorter yet correct responses (Hou et al., 2025; Su & Cardie, 2025; Aggarwal & Welleck, 2025). Specifically, if adopting a monotonically decreasing length reward function $f(l) = -\gamma l$ accepting the sequence length $l$ as input, the combined reward is defined as:

$$r'(o_i) = \begin{cases} r(o_i) - \gamma|o_i| & o_i \text{ is correct} \\ r(o_i) & \text{otherwise} \end{cases} \tag{4}$$

where $\gamma$ is a small factor to prevent the length reward from leading the overall reward, which could be adaptively computed (Zhang et al., 2025d) or preset as a hyperparameter (Kimi et al., 2025).

## 3 ON THE LIMITATIONS OF LENGTH-GUIDED REASONING OPTIMIZATION

In this section, we formally reveal two significant limitations of current length-reward driven efficiency reasoning under the representative critic-free RLVR algorithm, GRPO, by analyzing the

misalignment between the trajectory-level advantage score and the token-level optimization objective for redundant thinking tokens. Through an analysis of logit dynamics, we demonstrate that this misalignment degrades reasoning performance (§3.2) and fails to reduce early redundancies, thereby limiting potential gains in efficiency (§3.3). The concepts for each involved notation and abbreviation are illustrated in Table 4.

## 3.1 Logit Dynamics under Policy Gradient

The LRM policy at step $m$, as a softmax policy, is parameterized by

$$\pi_\theta^m(o_t \mid \boldsymbol{o}_{<t}) = \frac{\exp(z_{\boldsymbol{o}_{<t}, o_t})}{\sum_{o' \in |V|} \exp z_{\boldsymbol{o}_{<t}, o'}}, \tag{5}$$

where $z_{\boldsymbol{o}_{<t}, o_t}$ is the output logit of token $o_t$ given prefix $\boldsymbol{o}_{<t}$ and $o_t \sim \pi_\theta^m(\cdot \mid \boldsymbol{o}_{<t})$. Under the learning objective of the policy gradient, we have the following lemma (Cui et al., 2025):

**Lemma 1 (Difference of policy logits in vanilla policy gradient).** *Let the actor policy $\pi_\theta$ be a tabular softmax policy and updated using Eq. 1 with a learning rate $\eta$, the difference of $z_{\boldsymbol{o}_{<t}, o_t}$ between two consecutive steps $m$ and $m+1$ satisfies*

$$z_{\boldsymbol{o}_{<t}, o_t}^{m+1} - z_{\boldsymbol{o}_{<t}, o_t}^m = \eta \cdot \pi_\theta(o_t \mid \boldsymbol{o}_{<t}) \cdot A(\boldsymbol{o}_{<t}, o_t)$$

## 3.2 Optimization with Ill-posed Efficiency

GRPO estimates an advantage with intra-group relative reward by sampling $G$ rollouts repeatedly for a prompt. When $G$ rollouts contain both correct and incorrect trajectories, correct sequences always receive positive advantages, differing only in their magnitude and contributing little to efficiency optimization. In contrast, when rollouts generated by the policy $\pi_\theta$ on an easy prompt $q_{\theta, G}$ are all correct, the correctness reward becomes constant across trajectories, leaving length as the sole discriminative signal. As a result, correct yet overlong trajectories receive negative advantage estimates through the GRPO algorithm, which activates efficiency optimization.

Recently, Wang et al. (2025b) observes that the superior performance of LRMs is driven by high-entropy tokens, which lead the policy to conduct exploration and reflection. However, trajectory-level negative advantages would back-propagate to all tokens in Eq. 3, including the essential high-entropy tokens. Under Lemma 1, the negative advantages will cause the decline of probability for generating high-entropy tokens, and thereby the optimization process shifts from its intended goal, i.e., improving efficiency while preserving performance, to one that trades correctness for shorter trajectories. Formally, we could derive the following lemma:

**Lemma 2 (Decreased logits for correct high-entropy tokens).** *(Proof in Appendix A.1) For $f$ defined in Eq. 4, the expected change of logit for high-entropy tokens $\{o_{\text{high}}\}$ from $G$ correct rollouts $\{\boldsymbol{o}_i\}_{i=1}^G \sim \pi_\theta(\cdot \mid q_{\theta, G})$ sampled from $q_{\theta, G}$ between two consecutive optimization steps $m$ and $m+1$, is strictly negative:*

$$\mathbb{E}_{o \in \{o_{\text{high}}\}} \left[ z_o^{m+1} - z_o^m \right] < 0$$

In the above lemma, the correctly generated high-entropy tokens produced by $q_{\theta, G}$ have their generation probabilities reduced, which may disrupt or even distort the learning direction of an entire batch with respect to high-entropy tokens, subject to the constraints specified by the following theorem:

**Theorem 1 (Maintenance of High-entropy Tokens Under Batch Learning).** *(Proof in Appendix A.2) Let the ratio of prompts $q_{\theta, G}$ be $\kappa$. Assume that the length reward is defined as Eq. 4 and $\sigma_L$ is the standard deviation of response lengths of $q_{\theta, G}$ on average, the condition for which the expected logit change for correct high-entropy tokens among a batch is greater than 0 is as follows:*

$$\kappa \sigma_L < C,$$

*where $C$ is a constant with respect to the rollout tokens generated during a mini-batch.*

This theorem implies the condition under which the policy would suffer from performance degradation when applying length reward with GRPO. When $\kappa \sigma_L$ becomes too large, the policy no longer follows the performance-efficiency trade-off frontier. Instead, it shifts into a regime where gains in efficiency come at the cost of the proactivity of high-entropy tokens, thereby degrading performance.

### 3.3 Insufficient Efficiency

In addition to the decreased performance, current length-based reward methods also fail to achieve sufficient reduction of overthinking. Specifically, we differentiate the redundant tokens to be reduced by formally defining the necessary reasoning prefix as the most compact thinking process that supports deriving a correct answer for the first time:

**Definition 1** (**Necessary Reasoning Prefix**). *Let $q$ be an input prompt, $y^*$ be the ground truth answer, and $\boldsymbol{o} = (o_1, o_2, \ldots, o_L)$ be a generated response sequence on $q$, where $L = |\boldsymbol{o}|$. The necessary reasoning prefix (NRP) of $\boldsymbol{o}$ with respect to $q$ is the shortest prefix $\boldsymbol{o}_{1:K^*}$ such that* $\text{ANSWER}(\boldsymbol{o}_{1:K^*}) = y^*$ *and* $\forall k < K^*$, *either* $\text{ANSWER}(o_{1:k}) = \text{null}$ *or* $\text{ANSWER}(o_{1:k}) \neq y^*$.

As the correct answer is logically justified at position $K^*$, the token set $\{o_j \mid j > K_{\boldsymbol{o}}^*\}$ is considered **redundant** by many works (Dai et al., 2025; Yue et al., 2025). To prohibit the policy from continually generating further tokens after the already generated NRP tokens, we convert the objective to minimizing the probability of generating the first thinking token after the NRP, which functions on the reduction of holistic redundancy due to the autoregressive generation of LRMs:

$$\min \mathbb{E}_{\boldsymbol{o} \sim \pi_\theta(\cdot | q_{\theta,G})} \left[ z^m_{\boldsymbol{o}_{\leq K^*}, o_j} - z^{m-1}_{\boldsymbol{o}_{\leq K^*}, o_j} \right] \quad s.t. \quad j = K^* + 1 \tag{6}$$

Applying Lemma 1, this objective could be converted into a policy weighted expectation of advantages, which is shown to be positive:

**Theorem 2** (**Suboptimal Reduction of Redundant Tokens**). *(Proof in Appendix A.3) Let the reward function $f$ be defined as Eq. 4. Let $j = K_{\boldsymbol{o}}^* + 1$ denote the position of the first redundant token beyond the NRP in a correct rollout $\boldsymbol{o}$. Let $A(\boldsymbol{o})$ be the group-relative advantage computed via Eq. 2. Then, the expected policy gradient signal for the first overthinking token, denoted as $\mathcal{J}(A; j = K^* + 1) = \mathbb{E}_{\boldsymbol{o} \sim \pi_\theta(\cdot | q_{\theta,G})} [\pi_\theta(o_j \mid \boldsymbol{o}_{<j}) A(\boldsymbol{o}) \mid j = K_{\boldsymbol{o}}^* + 1]$ satisfies:*

$$\mathcal{J}(A; j = K^* + 1) > 0$$

This theorem tells us that although the policy would reduce thinking length by penalizing tokens far from the end of NRP from overlong responses, the policy cannot learn to stop at the end of NRP given no penalization on the first redundant token. This undesired property keeps partial overthinking tokens, leading to suboptimal reduction of redundancies.

## 4 DECS

Given the above analysis, we propose DECS, which contains three main designs to achieve the highest efficacy-efficiency tradeoff. First, to ensure that redundant tokens are penalized deterministically, we train a small module that precisely identifies necessary reasoning prefix (NRP) components for correct trajectories (§4.1). After that, we design a decoupled token-level reward and differentiate the reward scale for necessary and redundant tokens, to ensure enhanced efficiency without compromising performance (§4.2). Based on the conception of NRP, we propose to prevent aggressive penalization on high-entropy tokens following NRP by refactoring the data distribution of a batch according to the current levels of redundancy incrementally (§4.3). Fig. 2 illustrates the overall algorithm.

### 4.1 Detection of NRP

It is common practice to train a token-level classification model to annotate NRP components. However, it requires the same tokenizer as the policy, which hinders adaptation to other policies. To this end, we implement this detector as a lightweight generator model $\mathcal{M}_{\text{judge}}$, determining whether a reasoning chunk contains the correct answer to a given problem. Specifically, given a correct rollout $\boldsymbol{o}$, we first extract the reasoning tokens as $\boldsymbol{o}_{\text{think}} = (o_1, \ldots, o_{</\text{think}>})$. Using pre-defined separator tokens, the reasoning process is segmented into multiple chunks: $S = \{s_1, s_2, \cdots, s_{|S|}\}$, where $s_c$ is the $c$-th chunk of $\boldsymbol{o}_{\text{think}}$. The judgment $j_{s_c} \in \{\text{yes}, \text{no}\}$ for substep $s_c$ is generated by prompting $\mathcal{M}_{\text{judge}}$ given the problem $q$ and corresponding ground truth $y^*$ as:

$$j_{s_c} \sim \mathcal{M}_{\text{judge}}(\cdot \mid q, s_c, y^*) \tag{7}$$

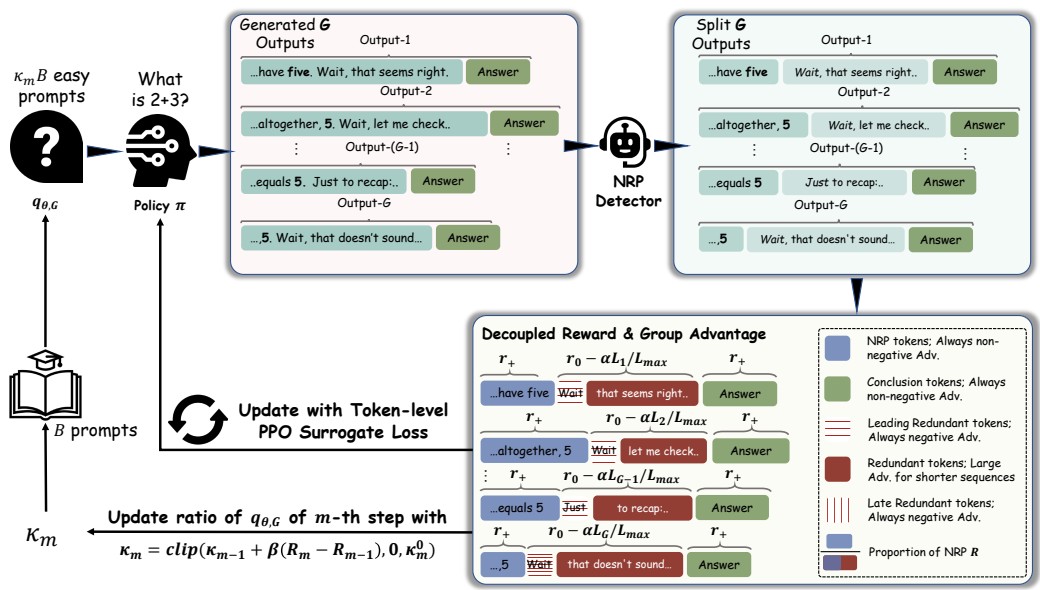

Figure 2: Overview of the DECS training pipeline. (1) **Decoupled Token-level Reward**: We fine-tune a small language model to detect the necessary reasoning prefix (NRP) from other redundancy, which are separately rewarded to penalize overthinking consistently while maintaining the probability for generating necessary reasoning steps. As the running example "What is 2+3?" shows, the NRP contains the reasoning chunks from the starting token to the first time the model generates the correct answer "5". After that, any leading redundant token like "Wait" receives negative advantages, and thereby discourage any redundant tokens to be generated via autoregressive generation. $\alpha = r_+ - r_0$. (2) **Curriculum Prompt Schedule**: The number of easy prompts $q_{\theta,G}$ grows in step with the progressive decline in remaining redundancy.

The NRP spans all reasoning chunks from the start through the first chunk whose judgment is "yes":

$$\text{NRP} = \bigoplus_{i=1}^{c^*} s_i, \quad \text{where } c^* = \min\{c \in [1, |S|] : j_{s_c} = \text{yes}\} \tag{8}$$

Here, $\bigoplus$ denotes the concatenation of reasoning chunks, and the $c^*$-th reasoning chunk is the first to entail the correct answer $y^*$. The training details are illustrated in Appendix G.6.

## 4.2 DECOUPLED REWARD ASSIGNMENT

For a group of rollouts $\{o_i\}_{i=1}^G$ generated based on a given prompt $q$, we design a token-level reward which ensures a maximum reward for NRP tokens and preferences for short yet correct responses:

$$r_{i,j} = \begin{cases} r_+ \cdot \mathbf{1}_{o_i \text{ is correct}} & j \leq K^*_{o_i} \vee o_j \notin o_{\text{think}} \vee o_{i,j} = \emptyset \\ (r_0 - \frac{(r_+ - r_0)L_i}{L_{\max}}) \cdot \mathbf{1}_{o_i \text{ is correct}} & j > K^*_{o_i} \wedge o_j \in o_{\text{think}} \end{cases} \tag{9}$$

where $r_+$ and $r_0$ are respectively the maximum and minimum positive rewards, $K^*_{o_i}$ is the last NRP token index of $o_i$ and $\emptyset$ denotes a padded token. Since the inverse proportional function enforces a far lower reward for redundant tokens compared to $r_+$, any token followed by the NRP would consistently receive negative advantages. Such penalization, as a result, helps to reduce verbosity via the autoregressive generation property of LRM regardless of sequence lengths. Besides, only redundant thinking tokens are possible to receive negative advantages, which prevents biased penalty on essential reasoning tokens and answer conclusion tokens, and sustains the policy during the RL training. Finally, the token-level advantage is computed similarly to GRPO and updated with Eq. 3:

$$A_{i,j}^{\text{DECS}} = \frac{r_{i,j} - \text{mean}(r_{1,j}, \ldots, r_{G,j})}{\text{std}(r_{1,j}, \ldots, r_{G,j})}. \tag{10}$$

Appendix C in detail explains the functionality of Eq. 9 for penalizing any leading redundant tokens.

### 4.3 CURRICULUM PROMPT SCHEDULE

After identifying NRP tokens, penalization of high-entropy tokens occurs only in redundant tokens following the NRP. Therefore, we schedule $\kappa_m$ based on the proportion of NRP $\mathcal{R}_m$ in correct sequences within a batch, which reflects how many correct high-entropy tokens would be penalized:

$$\kappa_m = \text{clip}(\kappa_{m-1} + \beta(\mathcal{R}_m - \mathcal{R}_{m-1}), 0, \kappa_m^0) \tag{11}$$

where $\kappa_m^0$ is the ratio of $q_{\theta,G}$ among the current sampled batch and $\beta$ is a hyperparameter to control the learning progress. As trajectories with zero advantages would not provide any learning signal, we follow Yu et al. (2025) to filter prompts whose $G$ rollouts are all incorrect and fill the batch by over-sampling. This curriculum strategy, designed to be bounded and monotonic, enables smooth adjustment in response to the observed NRP ratio, which aligns with the principle of curriculum learning (Bengio et al., 2009). By setting a moderate value $\beta$ with grid search (see Appendix H.1), DECS can satisfy the condition elucidated in Theorem 1 to maintain unbiased learning of high-entropy tokens throughout the whole training process. This yields stable convergence with no observed training instability or performance degradation, which is reflected in Fig. 6 and 7.

## 5 EXPERIMENTS

### 5.1 EXPERIMENT SETUPS

**Evaluation**  We use MATH500 (Lightman et al., 2023), AMC23 (AI-MO, 2024), Olympiad-Bench (He et al., 2024), AIME2024 (Mathematical Association of America, 2025a) and AIME2025 (Mathematical Association of America, 2025b) as in-domain testbeds, GPQA-Diamond (GPQA-D; Rein et al. (2024)) and LiveCodeBench-v6 (LCB; Jain et al. (2025)) as held-out testbeds, covering math, coding, and science tasks with diverse complexity. We choose **ThinkPrune** (Hou et al., 2025), **TLMRE** (Arora & Zanette, 2025), **AdaptThink** (Zhang et al., 2025b), **LC-R1** (Cheng et al., 2025) as baselines, and also include GRPO to serve as a performance reference. For fair comparison, we set the temperature as 0.6, top_p as 0.95, and use a maximum token limit of 16384 suggested by Guo et al. (2025). We conduct 128 rollouts for AIME2024, AIME2025, and AMC23, 16 rollouts for OlympiadBench, MATH500 and GPQA-D, and 10 rollouts for LCB to compute pass@1. We also compute the Average Efficiency Score (AES; Luo et al. (2025a)) for a comprehensive assessment of efficiency and efficacy. The details of both metrics are presented in Appendix G.4.

**Training**  We adopt DeepScaleR (Luo et al., 2025b) as the training set and choose DeepSeek-R1-Distill-1.5B (DS-1.5B), DeepSeek-R1-Distill-7B (DS-7B) as base policies. We perform 16 rollouts per prompt and use veRL (Sheng et al., 2025) as the training framework. $r_+, r_0$ in Eq. 9 are set to 1.1 and 1.0, respectively, while $\beta$ in Eq. 11 is set to 0.2 with grid-search. Additional hyperparameters are presented in Table 8.

### 5.2 RESULTS

As shown in Table 1, DECS reduces average reasoning length by 57.17% on the 1.5B model while improving pass@1 accuracy by +2.48 points, demonstrating simultaneous gains in efficiency and performance. On the 7B model, which exhibits less overthinking, DECS still cuts thinking tokens by 49.50%, outperforming all baselines, with a +0.8 point accuracy gain. Compared to the previous best, DECS improves the AES score by 0.12 and 0.14 on the 1.5B and 7B backbones, respectively, establishing a superior efficiency-performance trade-off that compresses the computation without sacrificing output quality. Meanwhile, although the NRP detector is specialized for math reasoning and the training data only cover the math corpus, such superiority of efficiency generalizes robustly to out-of-domain tasks (56.33% fewer tokens in GPQA-D and 33.52% fewer tokens in LCB), confirming DECS 's strong transferability and practical value for broader reasoning tasks.

### 5.3 ABLATION STUDY

In this section, we conduct an ablation study on the DS-1.5B base policy, to reveal the critical complementary relationship between the schedule prompt scheduling (CS) and decoupled token-level reward (DR). We show the results in Table 3 and plot the comparison in Fig. 3a. We observe

Table 1: Pass@1 (Acc) and the number of tokens (#Tok.) used across seven benchmarks. "LCB." denotes LiveCodeBench-v6, "OlympiadB." denotes the OlympiadBench, and "GPQA-D" denotes GPQA-Diamond. The best performing score is marked in **bold** and the second-best is underlined.

| Model | AIME2024 | | AIME2025 | | AMC23 | | MATH500 | | OlympiadB | | GPQA-D | | LCB | | Average | | |
|---|---|---|---|---|---|---|---|---|---|---|---|---|---|---|---|---|---|
| | Acc | #Tok. | Acc | #Tok. | Acc | #Tok. | Acc | #Tok. | Acc | #Tok. | Acc | #Tok. | Acc | #Tok. | Acc | #Tok. | AES |
| *DS-1.5B* | | | | | | | | | | | | | | | | | |
| Base | 27.99 | 12202 | 22.94 | 12138 | 69.84 | 7875 | 84.55 | 4847 | 53.78 | 9217 | 32.86 | 8540 | 24.53 | 10560 | 45.21 | 9340 | 0.00 |
| +GRPO | 32.76 | 8834 | 25.91 | 8431 | 77.09 | 5722 | 87.34 | 3577 | 58.73 | 6425 | 35.76 | 5953 | 26.45 | 8759 | 49.15 | 6814 | 0.53 |
| AdaptThink | 27.92 | 6914 | 21.95 | 7400 | 64.73 | 2644 | 81.57 | 1488 | 50.40 | 3501 | 25.92 | 4093 | 26.98 | 9181 | 42.78 | 5031 | 0.19 |
| ThinkPrune | 26.93 | 5306 | 20.86 | 4937 | 72.87 | 2869 | 84.27 | 1879 | 55.04 | 3477 | 35.51 | 3839 | 25.36 | 5515 | 45.83 | 3975 | 0.62 |
| TLMRE | 29.87 | 7550 | 22.24 | 7151 | 74.51 | 3943 | 84.86 | 2376 | 56.08 | 4833 | 33.74 | 4896 | 26.13 | 7737 | 46.78 | 5498 | 0.52 |
| LC-R1 | 23.65 | 6904 | 19.64 | 6681 | 68.69 | 3715 | 82.02 | 2277 | 51.57 | 4519 | 30.93 | 5377 | 23.54 | 6940 | 42.86 | 5202 | 0.18 |
| DECS | 31.25 | 5550 | 23.78 | 4965 | 75.37 | 2988 | 84.40 | 1817 | 56.10 | 3396 | 35.92 | 3255 | 27.66 | 6026 | 47.78 | 4000 | 0.74 |
| *DS-7B* | | | | | | | | | | | | | | | | | |
| Base | 50.65 | 10508 | 36.67 | 11096 | 88.77 | 5764 | 93.25 | 3654 | 69.22 | 7507 | 46.46 | 7502 | 45.95 | 8966 | 61.57 | 7857 | 0.00 |
| +GRPO | 52.50 | 9011 | 38.54 | 9670 | 91.88 | 5205 | 94.21 | 3520 | 72.59 | 6425 | 49.62 | 6101 | 47.71 | 8569 | 63.86 | 6929 | 0.23 |
| AdaptThink | 53.31 | 8884 | 36.48 | 9525 | 86.66 | 3675 | 91.06 | 1824 | 67.98 | 5528 | 43.91 | 5746 | 47.09 | 8209 | 60.93 | 6199 | 0.16 |
| ThinkPrune | 51.15 | 6625 | 36.46 | 7127 | 88.28 | 3193 | 92.98 | 2105 | 70.03 | 4154 | 48.42 | 4498 | 47.90 | 6881 | 62.17 | 4940 | 0.40 |
| TLMRE | 50.11 | 7023 | 34.24 | 7501 | 87.07 | 3329 | 91.83 | 2073 | 68.84 | 4382 | 49.02 | 4913 | 47.03 | 6772 | 61.16 | 5142 | 0.31 |
| LC-R1 | 50.52 | 6891 | 32.50 | 7387 | 85.74 | 2802 | 90.28 | 1473 | 67.76 | 3983 | 48.58 | 4672 | 46.83 | 6554 | 60.32 | 4823 | 0.28 |
| DECS | 51.33 | 5277 | 36.43 | 5516 | 89.04 | 2772 | 92.96 | 1728 | 70.28 | 3283 | 49.27 | 3276 | 48.05 | 5921 | 62.48 | 3968 | 0.54 |

Table 2: Generalization to the Qwen3-4B model. DECS still achieves 0.61 AES score, with 54.80% reduction to overthinking and 1.32 pass@1 improvement.

| Model | AIME2024 | | AIME2025 | | AMC23 | | MATH500 | | OlympiadB | | GPQA-D | | LCB | | Average | | |
|---|---|---|---|---|---|---|---|---|---|---|---|---|---|---|---|---|---|
| | Acc | #Tok. | Acc | #Tok. | Acc | #Tok. | Acc | #Tok. | Acc | #Tok. | Acc | #Tok. | Acc | #Tok. | Acc | #Tok. | AES |
| Qwen3-4B | 64.82 | 11611 | 56.30 | 12870 | 91.60 | 7478 | 93.74 | 4839 | 71.07 | 9144 | 39.11 | 8072 | 62.17 | 9713 | 68.40 | 9104 | 0.00 |
| DECS | 65.38 | 5431 | 56.96 | 5758 | 93.59 | 2864 | 93.78 | 1648 | 74.09 | 3646 | 41.00 | 3260 | 63.27 | 6196 | 69.72 | 4115 | 0.61 |

that without adaptive scheduling of easy problems, there is a noticeable performance drop, which verifies the impacts elucidated in Theorem 1. Meanwhile, without decoupled rewards, the policy remains nearly 25% of overthinking tokens, verifying that the sequence-level length reward fails to fully reduce overthinking as Theorem 2 implies.

## 5.4 BACKBONE GENERALIZATION

In this section, we generalize DECS to Qwen3 backbone model, where we apply DECS to Qwen3-4B (Yang et al., 2025) with the same training hyperparameters introduced in §5.1. Results in Table 2 demonstrates that DECS successfully extends to Qwen3-4B, with 54.80% reduction of reasoning tokens and 1.32 pass@1 improvement on average. This strongly implies that DECS is backbone-robust, and remains effective on a stronger base model.

## 6 ANALYSIS

In this section, we discuss the following research questions:

**RQ1:** How do the decoupled rewards help DECS to achieve the highest efficiency?
**RQ2:** How can DECS balance the exploration and exploitaiton when compressing reasoning?
**RQ3:** How does DECS perform with variable token budget?
**RQ4:** How do representative high-entropy tokens distribute after applying DECS?
**RQ5:** How does compressed thinking spread over various difficulty levels?

**Response to RQ1: Most of the tokens reduced by DECS stem from non-NRP tokens.** To reveal the significance of decoupled learning for reducing redundancy, we compute the proportion of NRP tokens in all thinking tokens (PNRP) of correct trajectories generated on AIME2024. We plot the average token costs and the average PNRP score in Fig. 3b. Although ThinkPrune reduces a similar number of thinking tokens as DECS, it achieves a relatively lower PNRP score. This inconsistency reflects that part of the reduced tokens stems from necessary reasoning tokens that contribute to the final correctness, which explains its performance drops in Table 1. Compared to LC-R1 remaining $\sim 10\%$ redundancy, DECS further reduces non-NRP tokens and improves the PNRP score, highlighting the utility of the decoupled reward for a unified reduction of overthinking.

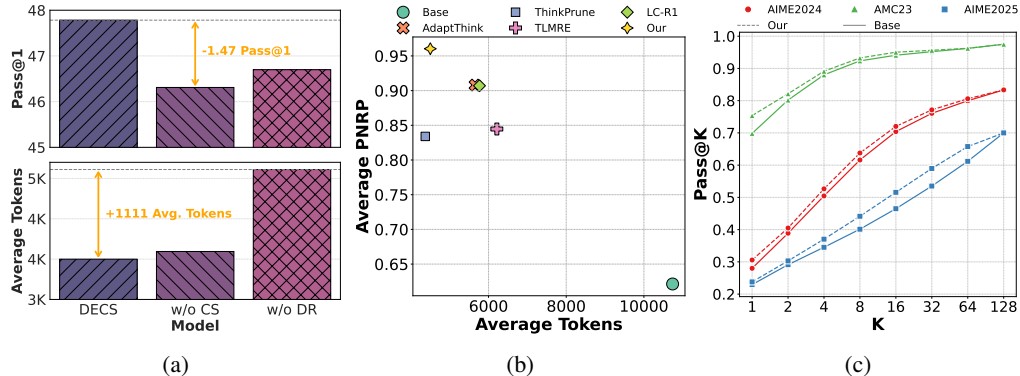

(a)  (b)  (c)

Figure 3: (a) Ablation study with two major components of DECS on the DS-1.5B base model. (b) Comparison of DECS with other baselines on the proportion of NRP tokens (PRNP) and actual reasoning tokens in the AIME2024 testbed. (c) DECS performs on par with the base policy (DS-1.5B) in terms of Pass@K scores on three challenging benchmarks.

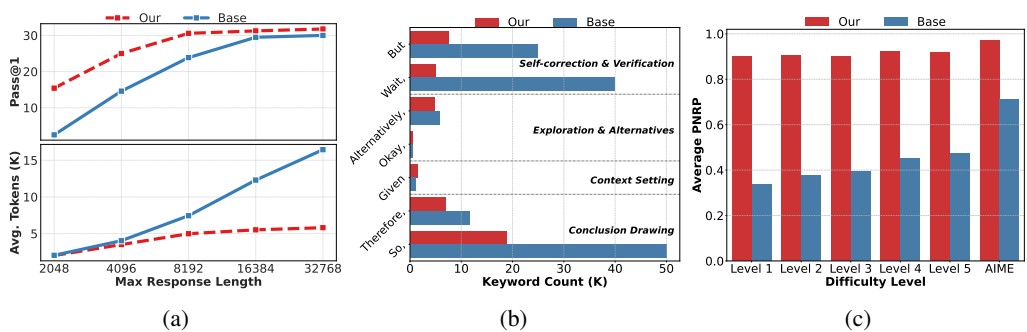

(a)  (b)  (c)

Figure 4: (a) Average tokens and Pass@1 performance with 5 increasing generation budgets; (b) Frequency of reasoning behavior tokens after applying DECS; (c) Consistent compression rates of DECS on six difficulty levels sourced from MATH500 and AIME2024.

**Response to RQ2: DECS maintains similar exploration potentials as the base model.** To investigate whether DECS achieves good pass@1-efficiency tradeoffs by sacrificing the problem-solving potentials compared to base models, we compare the pass@k scores ($k = \{2, 4, 8, 16, 32, 64, 128\}$) on AIME2024, AIME2025 and AMC23. Results in Fig. 3c and Fig. 8c reveal that across nearly all sample numbers, the success rate on the performance curve of the model compressed by our method almost perfectly overlaps with that of the original model. This result strongly demonstrates that the model's exploration ability to find a correct solution through multiple attempts is not injured by DECS. It suggests that preventing high-entropy tokens from receiving negative gradients sufficiently preserves most exploratory and creative properties.

**Response to RQ3: DECS consistently improves the token efficiency across diverse token budgets.** To validate whether the protection of NRP and exploratory high-entropy tokens would both improve the model's performance on token-constrained scenarios and not impair its performance with a less-constrained token limit (Snell et al., 2025), we evaluate under 5 increasing token limits: [2,048, 4,096, 8,192, 16,384, 32,768]. Fig. 4a, 8a, and 8b demonstrate the Pass@1 scores and average token costs on AIME2024, AIME2025 and AMC23 with the 1.5B policy. After applying DECS, the policy could use far fewer tokens to achieve competitive performance across diverse token limits, which holds even for the 32,768 context limit. For the 7B policy (depicted in Fig. 10), DECS performs on par with the base model with a negligible performance gap under the 32,768 token limit, but consuming fewer than 30% tokens. This further validates that DECS achieves superior efficiency compression without sacrificing the model's capability boundary excessively.

**Response to RQ4:   DECS reduces unnecessary reflective and conclusion tokens, but remains a consistent tendency for creative and context formulation tokens.**   To investigate how DECS refines the reasoning process and modulate the distribution for high-entropy decisive tokens, we analyzed the frequency of representative tokens with different reasoning behaviors, including "Self-Correction & Verification", "Exploration & Alternatives", "Context Setting" and "Conclusion Drawing" (Wu et al., 2025). Results in Fig. 4b reveal a significant shift in the tendency for correction tokens with DECS, which is the main source of overthinking. Meanwhile, the negligible change in the frequency of exploratory tokens also suggests that the shearing of tokens after NRP hardly cause degradation of creative thinking. Also, the dramatic decrease of conclusion tokens reflects that after applying DECS the policy is more confident in their reasoning intermediate results, which leads to similar or even higher pass@1 scores across diverse benchmarks.

**Response to RQ5:   DECS compresses non-NRP tokens under variable input complexity.** Since large reasoning models (LRMs) often overthink even on easy queries, we examine whether DECS consistently reduces overthinking across varying difficulty levels. We compute the PNRP score on the MATH500 and AIME2024 datasets, which provide self-contained difficulty gradients across six levels. As shown in Fig. 4c, PNRP scores increase with problem difficulty, and DECS consistently achieves scores above 90% across all levels. This trend holds for the 7B model (Fig. 9b), with even higher scores on AIME2024, likely due to its improved reasoning ability and reduced inherent overthinking. These results confirm that DECS enhances reasoning efficiency in LRMs across diverse inputs, demonstrating its effectiveness and generality.

## 7 CONCLUSION

In this paper, we theoretically identify two key flaws in current critic-free RL methods for reducing the overthinking phenomenon, which stems from the misalignment between token-level overthinking reduction and sequence-level reward assignment. To mitigate these two drawbacks, we propose DECS, which proposes a decoupled reward system for NRP and non-NRP tokens for consistent reduction of overthinking, and introduces curriculum batch scheduling for maintaining exploratory potentials. Experiments show that DECS reduces $\sim$50% of reasoning tokens while maintaining or improving performance, achieving more efficient reasoning without sacrificing accuracy.

## ACKNOWLEDGMENTS

We thank the anonymous reviewers for their insightful comments and suggestions. This work was supported by the National Key R&D Program of China (No. 2022ZD0162101), National Natural Science Foundation of China (No. 62576209) and STCSM (No. 2025SHZDZX025G05).

## 8 REPRODUCIBILITY STATEMENT

In this section, we list any related materials that help to reproduce this paper

1. **Datasets**: The training set we used is described in §5.1 and evaluation sets we used are described in Appendix G.2.

2. **Theoretical Support**: Any assumptions, lemmas, propositions, theorems and corresponding proofs are detailed in Appendix A.

3. **Code**: The code to reproduce our algorithm would be put into the supplementary materials.

4. **Computational Resources**: We use 4xNVIDIA A100 80GB GPUs to conduct all experiments.

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

# A  THEORETICAL SUPPORT

## A.1  PROOF OF LEMMA 2

**Lemma 2:** *For $q_{\theta,G}$ defined as above and $f$ defined in Eq. 4, for any correct token that belongs to the high-entropy token defined in Wang et al. (2025b), the expected change of logit for high-entropy tokens $\{o_{\text{high}}\}$ from $G$ correct rollouts $\{o_i\}_{i=1}^{G} \sim \pi_\theta(\cdot \mid q_{\theta,G})$ between two consecutive optimization steps $m$ and $m+1$ is strictly negative: :*

$$\mathbb{E}_{o \in \{o_{\text{high}}\}} \left[ z_o^{m+1} - z_o^m \right] < 0$$

*Proof.* We assume that the high-entropy token is distributed uniformly. For any sequence with $L_i$ tokens, there will be $h \cdot L_i$ high-entropy tokens on average, where $h$ is defined as $20\%$ in Wang et al. (2025b). For simplicity, we denote $\mathbb{E}_{o \in \{o_{\text{high}}\}} \left[ z_o^{m+1} - z_o^m \right]$ as $\mathbb{E} \left[ z_{o_{\text{high}}}^{m+1} - z_{o_{\text{high}}}^m \right]$. Since the length

reward is a linear function *w.r.t.* length, the logit difference expectation is computed as:

$$
\mathbb{E}\left[z_{o_{\text{high}}}^{m+1} - z_{o_{\text{high}}}^{m}\right] \propto \sum_{i=1}^{G}\sum_{j=1}^{L_i} \pi_\theta(o_{i,j}) \cdot A(o_{i,j}) \cdot \mathbf{1}_{o_{i,j}\in\{o_{\text{high}}\}}
$$

$$
= \sum_{i=1}^{G} A(o_i) \underbrace{\sum_{j=1}^{L_i} \pi_\theta(o_{i,j}) \cdot \mathbf{1}_{o_{i,j}\in\{o_{\text{high}}\}}}_{\text{GRPO Broadcast}}
$$

$$
< \sum_{i=1}^{G} A(o_i) \cdot hL_i
$$

$$
= h\sum_{i=1}^{G} \frac{f(L_i) - \mu_{f(L)}}{\sigma_{f(L)}} L_i
$$

$$
= \frac{hG}{\sigma_L} \cdot \underbrace{\frac{1}{G}\sum_{i=1}^{G} -L_i^2 + L_i\mathbb{E}[L]}_{\text{Linear transformation of length}}
$$

$$
= \frac{hG}{\sigma_L}(-\mathbb{E}[L^2] + \mathbb{E}[L]^2)
$$

$$
= -hG\sigma_L < 0
$$

As a result, $\mathbb{E}\left[z_{o_{\text{high}}}^{m+1} - z_{o_{\text{high}}}^{m}\right] < 0$ for prompt $q_{\theta,G}$. $\qquad\square$

## A.2 PROOF OF THEOREM 1

**Theorem 1:** *Let the ratio of prompts $q_{\theta,G}$ be $\kappa$. Assume that the length reward is defined as Eq. 4 and $\sigma_L$ is the standard deviation of response lengths of $q_{\theta,G}$ on average, the condition for which the expected logit change for correct high-entropy tokens among a batch is greater than 0 is as follows:*

$$
\kappa \cdot \sigma_L < C,
$$

where $C$ is a constant with respect to the rollout tokens generated during a mini-batch.

*Proof.* Following the proof process of Lemma 2, we can compute the upper bound of $\mathbb{E}_B\left[z_{\text{high}}^{m} - z_{\text{high}}^{m-1}\right]$ by computing the upper bound of $\sum_{n=1}^{B}\sum_{i=1}^{G}\sum_{j=1}^{|o_n^i|} A(o_n^i)\cdot\mathbf{1}_{\text{correct}}\cdot\mathbf{1}_{\text{high}}$, which is the sum of advantage values of all correct high-entropy tokens:

$$
\mathbb{E}_B\left[z_{\text{high}}^{m} - z_{\text{high}}^{m-1}\right] \propto \sum_{n=1}^{B}\sum_{i=1}^{G}\sum_{j=1}^{|o_n^i|} \pi_\theta(o_n^{i,j})A(o_n^{i,j}) \cdot \mathbf{1}_{\text{correct}} \cdot \mathbf{1}_{\text{high}} \tag{12}
$$

$$
< \sum_{n=1}^{B}\sum_{i=1}^{G} hL_i A(o_n^{i,j}) \cdot \mathbf{1}_{\text{correct}}
$$

where $o_n^{i,j}$ is the $j$-th token of $i$-th output within a group of responses generated based on $n$-th prompt. Assume that the output distribution for any $q_{\theta,G}$ is *i.i.d*, we expand this term as follows:

$$
\sum_{n=1}^{B}\sum_{i=1}^{G} hL_i A(o_n^{i,j}) \cdot \mathbf{1}_{\text{correct}} = \kappa Bh\sum_{i=1}^{G} L_i \cdot \frac{f(L_i) - \mu_{f(L)}}{\sigma_{f(L)}} + \sum_{n=1}^{\kappa B}\sum_{i=1}^{G} L_n^i \cdot \frac{1 - \mu_n^c}{\sigma_n^c} \cdot \mathbf{1}_{\text{correct}}
$$
$$
\tag{13}
$$

$$
= -\kappa BhG\sigma_L + \sum_{n=1}^{\kappa B}\sum_{i=1}^{G} L_n^i \cdot \frac{1 - \mu_n^c}{\sigma_n^c} \cdot \mathbf{1}_{\text{correct}}
$$

where $\mu_n^c, \sigma_n^c$ is the average and variance of the correctness reward for $n$-th prompt. Assume that the number of correct responses for each prompt $b$ is $a_n$, $\mu_n^c = a_n$ and $\sqrt{\sigma_n^c} = \sqrt{a_n(1-a_n)}$, we expand the second term as:

$$\sum_{n=1}^{\kappa B} \sum_{i=1}^{G} L_n^i \cdot \frac{1 - \mu_n^c}{\sigma_n^c} \cdot \mathbf{1}_{\text{correct}} = \sum_{n=1}^{\kappa B} \sum_{i=1}^{G} L_n^i \cdot \frac{1 - a_n}{\sqrt{a_n(1-a_n)}} \cdot \mathbf{1}_{\text{correct}} = C_B \qquad (14)$$

Therefore, the second term is always positive as $a_n < 1$ and is a constant within the batch $B$. As a result, the objective would be less than 0 if and only if the first term of Eq. 13 is sufficiently negative. Solving the inequality that Eq. 13 is positive, we obtain the following condition for which the learning signal across the batch would not penalize correct high-entropy tokens:

$$\kappa \cdot \sigma_L < \frac{C_B}{BhG} = C \qquad (15)$$

There are two ways to break this condition: (1) more $q_{\theta,G}$ within a prompt, parameterized as a larger $\kappa$; and (2) a larger range of output length, parameterized as larger values of $\sigma_L$.

$\square$

### A.3 PROOF OF THEOREM 2

**Theorem** 2 *Let the reward function $f$ be defined as Eq. 4. Let $j = K_o^* + 1$ denote the position of the first redundant token beyond the NRP in a correct rollout $o$. Let $A(o)$ be the group-relative advantage computed via Eq. 2. Then, the expected policy gradient signal for the first overthinking token, denoted as $\mathcal{J}(A; j = K^* + 1)\mathbb{E}_{o \sim \pi_\theta(\cdot|q_{\theta,G})}\left[\pi_\theta(o_j \mid o_{<j})A(o) \mid j = K_o^* + 1\right]$ satisfies:*
$$\mathcal{J}(A; j = K^* + 1) > 0$$

*Proof.* Considering an arbitrary easy prompt $q$, suppose the LRM $\pi_\theta$ generates a group of $G$ correct rollouts $\{o_i\}_{i=1}^{G}$ with lengths $L_1, L_2, \ldots, L_G$. The advantage for each rollout is computed via group-wise standardization (Eq. 2):

$$A(o_i) = \frac{f(L_i) - \mu_G}{\sigma_G} = -\gamma \frac{L_i - \bar{L}}{\sigma_L},$$

where $\bar{L}$ and $\sigma_L$ are the mean and standard deviation of $L$ within the group. Assume a simplified optimal reward function $A^{\text{opt}} = -\gamma$ for all $i = 1, 2, \ldots, G$. Now consider the first redundant token $o_{i,j=K_i^*+1}$ generated after the Necessary Reasoning Prefix (NRP) in rollout $o_i$. In GRPO, this token inherits the sequence-level advantage $A(o_i)$, and contributes to the policy gradient through the term:

$$\pi_\theta(o_j \mid o_{<j}) \cdot A(o_i).$$

Taking the expectation over the group:

$$\mathbb{E}_{o \sim \pi_\theta(\cdot|q)}\left[\pi_\theta(o_j \mid o_{<j})A(o) \mid j = K_o^* + 1\right].$$

Formally, Let $w_i := \pi_\theta(o_{i,\leq j} \mid q)$ be the probability of generating the prefix up to the first redundant token and simplify $\mathcal{J}(A; j = K^* + 1)$ as $\mathcal{J}(A)$. Under the typical behavior of autoregressive policies, shorter rollouts have higher generation probability: if $L_i < L_k$, then $w_i \geq w_k$.

We only consider the case where all rollouts are redundant, i.e., for any rollout $o_i$, its thinking length is larger than its NRP length $L_i > K_i^*$. By the definition of conditional expectation:

$$\mathcal{J}(A) = \frac{\sum_{i=1}^{G} w_i \cdot A(o_i)}{\sum_{i=1}^{G} w_i} = -\gamma \cdot \frac{\bar{L}_w - \bar{L}}{\sigma_L},$$

where $\bar{L}_w = \frac{\sum_{i=1}^{G} w_i L_i}{\sum_{i=1}^{G} w_i}$ is the policy weighted average length. Subtracting $\bar{L}_w$ with $\bar{L}$, we obtain:

$$\bar{L}_w - \bar{L} = \frac{\sum_{i=1}^{G} w_i L_i}{\sum_{i=1}^{G} w_i} - \sum_{i=1}^{G} L_i$$
$$= \frac{\sum_{i=1}^{G} w_i L_i - (\sum_{i=1}^{G} w_i)(\sum_{i=1}^{G} L_i)}{\sum_{i=1}^{G} w_i} < 0,$$

Therefore, $\bar{L}_w - \bar{L} < 0$, so $\mathcal{J}(A) > 0$.

$\square$

Table 3: Ablation study with two major components of DECS on the DS-1.5B base model. "CS" denotes adaptive data sampling and "DR" denotes the decoupled reward mechanism.

| Model | AIME2024 | | AIME2025 | | AMC23 | | MATH500 | | OlympiadB | | GPQA-D | | LCB | | Avg | |
|---|---|---|---|---|---|---|---|---|---|---|---|---|---|---|---|---|
| | Acc | #Tok. | Acc | #Tok. | Acc | #Tok. | Acc | #Tok. | Acc | #Tok. | Acc | #Tok. | Acc | #Tok. | Acc | #Tok. |
| DECS | **31.25** | **5550** | **23.78** | **4965** | **75.37** | **2988** | **84.40** | **1817** | **56.10** | **3396** | **35.92** | **3255** | **27.66** | **6026** | **47.78** | **4000** |
| w/o CS | 28.78 | 5676 | 23.28 | 5206 | 71.99 | 3125 | 83.77 | 1830 | 55.12 | 3477 | 34.44 | 3375 | 26.79 | 5972 | 46.31 | 4095 |
| w/o DR | 30.23 | 6942 | 22.60 | 6850 | 73.50 | 3548 | 83.94 | 1917 | 55.85 | 4513 | 33.71 | 4444 | 27.05 | 7561 | 46.70 | 5111 |

# B  RELATED WORK

In this section, we introduce three categories of work to improve reasoning efficiency with training:

**Methods based on length rewards**   As is introduced in the main text, length reward is one of the most influential strategies to improve efficiency for large reasoning models (LRM). Su & Cardie (2025) proposes to determine the reward scale coefficient $\gamma$ defined in Eq. 4 by comparing the average accuracy of the current step with that of the reference model. Aggarwal & Welleck (2025) proposes L1, which optimizes the LRM to generate a correct reasoning path within a certain context window. Lyu et al. (2025) also proposes to use length penalties to encourage the model to answer problems with different complexity with adaptive token limits. Cheng et al. (2025) uses a specialized model to separate the necessary reasoning part and the redundant tokens, and reinforce the policy to output the response with the shortest NRP length. However, although they have made great efforts to encourage correct yet short reasoning paths, they do not consider the internal logit dynamics when applying sequence-level length rewards with token-level optimization objectives, and thus suffer from performance degradation.

**Methods based on Adaptive Reasoning**   Another line of work is to teach the policy to directly enclose the reasoning process and directly output the answer for easy problems, while conducting sufficient reasoning for difficult queries. AdaptThink (Zhang et al., 2025b) proposes to generate a group of responses for a single prompt, half of which is generated with no thinking content, and thereby guides the policy for necessary thinking. Zhang et al. (2025d) combines the length reward and AdaptThink to teach the policy to conduct further thinking for already enclosed reasoning processes, to avoid the policy from outputting incorrect conclusions with insufficient reasoning. Zhang et al. (2025c) constructs a dataset containing both responses from thinking models and non-thinking models, respectively and proposes to fine-tune the policy to choose thinking modes according to the difficulty of problems. Although reducing the average output tokens successfully, these methods often erroneously adopt non-thinking modes for challenging problems, which leads to performance degradation. Meanwhile, they lack a penalty for overlong responses generated in thinking mode, thereby still remaining overthinking.

**Methods based on substep truncation**   As most overthinking contents could be separated into thinking chunks with some high-entropy tokens, some methods propose to truncate the redundant chunks following the NRP to achieve efficiency optimization. MinD (Zeng et al., 2025b) uses supervised fine-tuning to teach the policy to output its thinking contents with explicit separator tokens in a cold-start manner. After that, it uses GRPO to teach the policy to stop after generating the NRP part. S-GRPO (Dai et al., 2025) splits a full reasoning trace into multiple segments, and manually prompts the policy to derive the final answer with a different number of chunks so that the policy could output trajectories containing only the NRP part. Yue et al. (2025) shares a similar philosophy with S-GRPO, but proposes to assign process-level rewards after each split segment to encourage the policy to stop generation on the token with max cumulative rewards. However, most of these methods either introduce additional rollouts, or are not end-to-end frameworks, which hinders their actual adaptation to large-scale training or agentic applications (Zhang et al., 2025a).

# C  DETAILED ANALYSIS OF DECOUPLED REWARD DESIGN

While the decoupled reward formulation in Eq. 9 does not explicitly differentiate between *leading redundant tokens* (i.e., the first token immediately following the Necessary Reasoning Prefix, NRP) and other redundant tokens, the combination of this design with the group-relative advantage

Table 4: Concepts of involved mathematical notations, symbols and abbreviations.

| Symbol | Description |
|---|---|
| $\pi_\theta$ | The policy model parameterized by $\theta$. |
| $q$ | A given input prompt or question. |
| $\boldsymbol{o}$ | A reasoning trace (output sequence) generated by the policy $\pi_\theta$. |
| $\boldsymbol{o}_{<j}$ | The sub-sequence of the output $o$ up to (but not including) the $j$-th token. |
| $o_j$ | The $j$-th token in the output sequence $o$. |
| $r$ | The reward signal received from the verifier. |
| $\nabla J(\theta)$ | The gradient of the policy objective function with respect to parameters $\theta$. |
| $A(\boldsymbol{o}_{<j}, j)$ | The advantage function at step $j$ given the state $\boldsymbol{o}_{<j}$, indicating how much better the action is compared to average. |
| NRP | Necessary Reasoning Prefix, the minimal set of prefix tokens of a reasoning trace required to reach a correct answer. Formal definition in Definition 1. |
| PNRP | Proportion of the NRP tokens among a complete reasoning trace. |
| $L_i, |\boldsymbol{o}_i|$ | The total length of the $i$-th rollout (total number of tokens generated). |
| $\kappa_m$ | The proportion of easy prompts among a batch at iteration $m$ |
| $\beta$ | Coefficient for increasing the $\kappa$ value for each iteration. |
| $r_+$ | Maximum reward assigned to necessary reasoning tokens. |
| $r_0$ | Base reward value. |
| AES | Average Efficiency Score, a metric combining pass@1 score and token cost. |
| $\oplus$ | Concatenation operator for reasoning chunks. |
| $c^*$ | The index of the first reasoning chunk that entails the correct answer. |
| $K_{c^*}^i$ | The cumulative token count up to the end of the $c^*$-th chunk in the $i$-th rollout. |

mechanism in DECS ensures that all redundant tokens consistently receive negative advantages during training. This property arises from the interplay between the reward structure and the relative advantage computation:

1. Reward Structure: For any correct rollout $o_i$, tokens within the NRP receive a fixed high reward $r_+ = 1.1$. Tokens after the NRP (redundant tokens) receive a length-scaled reward:

$$r_{i,j} = r_0 - \frac{(r_+ - r_0)L_i}{L_{\max}}, \quad \text{where } L_i = |o_i|.$$

2. Group-Relative Advantage Mechanism: A token receives positive advantage only if its reward exceeds the average reward at that position across the group of $G$ rollouts.

Consider the leading redundant token in a sequence:

**Case 1: The current sequence has the longest NRP in the group.** Its reward computed by Eq. 9 is generally below-average the average reward among the token group. In this case, it receives negative advantages. This could be empirically verified across different training steps and model architectures. To be more specific, we evaluated rollouts from the DS-1.5B/DS-7B model at training steps [1, 60, 120, 180, 240] and counted how many leading redundant tokens have an above-average reward computed by Eq. 9 in Table 5. Results demonstrate that across all batches and positions, no instance was found where a leading redundant token received a positive advantage.

**Case 2: At least one sequence in the group has an equally long or longer NRP.** Then, there exists at least one token in that sequence receiving the full reward $r_+ = 1.1$. Meanwhile, under the condition where a longer sequence has longer NRP length, we could sort the 16 sequence lengths in descending order and assume that for a sequence length $L_k$ which is the $k-$th largest among the remaining 15 sequences. In this situation, there would be at least $k$ rewards that equal to 1.1 and the remaining rewards are all less than or equal to $r_k$. Therefore,

$$r_k - \mu = 1.0 - 0.1 \times L_k/L_{\max} - \frac{1}{16}(1.1 \times k + \sum_{i=1}^{16-k} 1.0 - 0.1 \times L_i/L_{\max})$$

Table 5: Comparison of erronously rewarded redundant tokens using Eq. 9 on DS-1.5B/DS-7B.

| Step | 1 | 60 | 120 | 180 | 240 |
|------|---|----|-----|-----|-----|
| DS-1.5B | 0 | 0 | 0 | 0 | N/A |
| DS-7B | 0 | 0 | 0 | 0 | 0 |

Table 6: Human evaluation results on the classification accuracy of math-specialized NRP detector on other domains, including science and coding.

| Dataset | Human1 | Human2 | Human3 |
|---------|--------|--------|--------|
| GPQA-D | 99.18 | 98.96 | 99.23 |
| LiveCodeBench | 97.88 | 97.63 | 97.49 |

Table 7: Timing consumption (seconds) for the NRP detector within a full training step.

| Step | 1 | 60 | 120 | 180 | 240 |
|------|------|------|------|------|------|
| **Total time** | 3791 | 3038 | 2972 | 2368 | 2583 |
| **NRP detect time** | 130 | 121 | 101 | 121 | 94 |
| **Ratio** | 0.034 | 0.040 | 0.034 | 0.051 | 0.036 |

where $\forall i, L_i \leq L_k$ by above conditions. Simplifying the right term, we could obtain:

$$
\begin{aligned}
r_k - \mu &\leq 1 - 0.1\frac{L_k}{L_{\max}} - 1 - 0.1\frac{k - (16-k)\frac{L_k}{L_{\max}}}{16} \\
&= -0.1\frac{k}{16}(1 + \frac{L_k}{L_{\max}}) \\
&< 0
\end{aligned}
$$

This holds for any leading redundant tokens for any sequence that does not have longest NRP length, which also represents a negative advantage value.

In both cases, the reward for the leading redundant token is strictly below the group average, resulting in a negative advantage. This guarantees it will be penalized by the policy gradient update. Thus, the decoupled reward design, in conjunction with the advantage estimator of GRPO, inherently penalizes leading redundant tokens, thereby penalizing all redundancies by autoregressiveness.

# D  CORRELATION TO RELATIVE OVERGENERALIZATION IN MULTI-AGENT RL

In this section, we discuss the parallels between the "erroneous penalization" issue presented in this paper and concepts like relative overgeneralization in Multi-Agent RL (MARL). The core problem, the **misalignment between global reward signals and local token-level updates**, resonates with broader challenges in the RL literature.

- **The Parallel**: Relative overgeneralization in MARL occurs when agents learn suboptimal joint behaviors due to misleading credit assignment from shared rewards. Analogously, in our single-agent sequence setting, the coarse-grained, sequence-level length penalty acts as a blunt signal. This signal fails to accurately attribute cost to specific redundant tokens, leading to the **erroneous suppression of useful, high-entropy tokens**. This is essentially a form of **token-level overgeneralization**.
- **The Solution**: This parallel highlights a fundamental challenge in policy gradient methods: the need for fine-grained, temporally precise feedback to avoid spurious credit assignment. Our proposed **decoupled reward mechanism** (Eq. 9) can be viewed as an instance of **structured credit assignment**. By explicitly disentangling necessary and redundant reasoning steps, we ensure that only truly redundant tokens are penalized, effectively mitigating this token-level overgeneralization.

Table 8: Hyperparameters for DᴇCS training.

| Hyperparameter | R1-Distill-Qwen-1.5B | R1-Distill-Qwen-7B |
|---|---|---|
| max response length | 16384 | 8192 |
| batch szie | 128 | 128 |
| rollout batch size | 128 | 128 |
| learning rate | 2.0e-06 | 2.0e-06 |
| total training epochs | 2 | 3 |
| rollout number | 16 | 16 |
| $\epsilon$ | 0.2 | 0.2 |
| $\beta$ | 0.2 | 0.2 |

## E  LIMITATION & FUTURE WORK

Our approach effectively mitigates the performance–efficiency trade-off in length-rewarded GRPO by dynamically separating necessary reasoning from redundant tokens. That said, two practical considerations remain.

First, the NRP detector is implemented as a small auxiliary model (1.5B parameters). While this adds a minor component to the pipeline, it incurs only 5.1% training overhead (Table 7) and achieves near-perfect accuracy (Fig. 6c), making it a lightweight and reliable proxy. Integrating NRP detection directly into the policy, e.g., via confidence (Yan et al., 2025) or entropy signals (Cui et al., 2025), is a promising future direction but not required for the current solution to work well.

Second, we evaluate DᴇCS on models up to 7B due to resource constraints. However, since our method is model-agnostic and controls learning solely through the curriculum schedule with Eq. 11, we expect it to scale smoothly to larger architectures with adequate compute.

Importantly, neither limitation affects the validity or effectiveness of our core contribution: a simple, low-overhead strategy that achieves strong efficiency gains without sacrificing performance across both in-domain and out-of-domain tasks.

## F  USE OF LARGE LANGUAGE MODELS

We mainly use large language models for proofreading and polishing of this paper.

## G  EXPERIMENTAL DETAILS

In this section, we provide the details of each experiment conducted throughout this paper. We provide detailed descriptions of the training hyperparameters, the test sets and prompts used for evaluation, the metrics employed to evaluate each method, and detailed procedures for reproducing important experiments.

### G.1  TRAINING HYPERPARAMETERS

We present the other hyperparameters adopted during training in Table 8. Since we schedule prompts during a batch in §4.3 and use over-sampling to complement a full batch, the number of total training steps is half-reduced. To align with a similar number of training updates with other baselines, we train the base model for 2 epochs for the 1.5B model. Note that we train one more epoch for the 7B base model, as we set a max response length to 8192 and hence many responses exceeding this limit would be filtered out. As a result, to achieve a similar number of prompts participating in the training process, we extend the training process by allowing for training one more epoch.

### G.2  DESCRIPTIONS OF TESTBEDS

We present the detailed description of the evaluation datasets as follows:

1. **AIME2024, AIME2025** (Mathematical Association of America, 2025a;b): These two datasets contain High school Olympiad-level assessment from American Invitational Mathematics Examination in 2024 and 2025. Each dataset contains 30 challenging problems covering Algebra/Geometry/Number theory.

2. **AMC23** (AI-MO, 2024): This dataset is sourced from American Mathematics Competitions system in 2023, which contains 40 problems with hybrid question types.

3. **OlympiadBench** (He et al., 2024): This dataset contains comprehensive math Olympiad problems from various nations. We only select the English version related to Math and keep the problems that require an answer with a number, leaving 581 problems for evaluation in total.

4. **MATH500** (Lightman et al., 2023): This dataset is an advanced mathematics evaluation set curated by OpenAI containing 500 problems with formal mathematical notations.

5. **GPQA-Diamond** (Rein et al., 2024): This dataset is a subset of the GPQA (Graduate-Level Google-Proof Q&A) dataset, which contains 198 challenging multiple-choice questions authored and verified by domain experts in biology, physics, and chemistry.

6. **LiveCodeBench** (Jain et al., 2025): This dataset is designed to evaluate the live code generation capabilities of large language models, focusing on immediate correctness and practical coding skills. We use its `v6` version, containing 1,055 problems in total.

### G.3    EVALUATION PROMPTS

For AIME2024, AIME2025, AMC23, OlympiadBench, and MATH500, we prompt the LRM with "`Please reason step by step and output the final answer within \boxed{}`" and use Math-Verify[1] to evaluate the correctness. For GPQA-Diamond, we prompt the LRM with "`Please reason step by step and put the answer index after ANSWER: `". For LiveCodeBench, we prompt the LRM with "`You will be given a question (problem specification) and will generate a correct Python program that matches the specification and passes all tests.\n\nQuestion: {question}\n\nRead the inputs from stdin solve the problem and write the answer to stdout (do not directly test on the sample inputs). Enclose your code within delimiters as follows. Ensure that when the python program runs, it reads the inputs, runs the algorithm and writes output to STDOUT.`"

### G.4    COMPUTATION OF METRICS

**AES**    The AES score (Luo et al., 2025a) is computed by comprehensively comparing the pass@1 score and average token costs of the tuned policy and the base policy.

$$\text{AES} = \frac{L_{\text{base}} - L}{L_{\text{base}}} + \begin{cases} 3 \cdot \frac{\text{pass@1} - \text{pass@1}_{\text{base}}}{\text{pass@1}_{\text{base}}} & \text{pass@1} \geq \text{pass@1}_{\text{base}} \\ -5 \cdot \frac{\text{pass@1}_{\text{base}} - \text{pass@1}}{\text{pass@1}_{\text{base}}} & \text{pass@1} < \text{pass@1}_{\text{base}} \end{cases} \tag{16}$$

This metric incorporates both the ratio of tokens reduced and the impact on model performance: it penalizes methods that degrade performance while rewarding those that improve upon the baseline.

**Pass@K**    The pass@K (Chen et al., 2021) scores are computed as below:

$$\text{pass@K} = 1 - \frac{\binom{n-c}{K}}{\binom{n}{K}} \tag{17}$$

where $n$ is the number of samples and $c$ is the number of correct samples. When $K$ is set to 1, this metric is reduced to the average accuracy among the $n$ samples.

### G.5    DETAILS OF EXPERIMENTS OF FIGURE 1

To compute the optimal curve, we use the results obtained in Fig. 4a to serve as the points to be fitted. Specifically, when setting the maximum context length to $[2048, 4096, 8192, 16384, 32768]$,

---

[1]https://github.com/huggingface/Math-Verify

we record the actual output tokens of DECS as $[2010, 3504, 4985, 5518, 5808]$, and corresponding pass@1 scores as $[15.42, 25.00, 30.57, 31.25, 31.77]$. After that, we use the well-established log-linear scaling law function $y = a \log_2 x + b$ (Muennighoff et al., 2025; Zeng et al., 2025a; Ballon et al., 2025) to fit these data points, and obtain the fitter function as $y = 0.1083 \log_2 x - 1.0306$ where $x$ represents the average tokens and $y$ represents the pass@1 score, with an $R^2 = 0.9936$ (Hastie et al., 2009). After that, we plot the base model's performance (labeled as 'Base') and LC-R1's performance (labeled as 'Previous Method') to show that previous length-penalty based methods fail to drive the policy towards the optimal trade-off between token efficiency and model expressiveness.

### G.6 DETAILS OF TRAINING NRP DETECTOR

**Training Details**   We build the training data using OpenR1-Math-220K[2] with the large language model Qwen2.5-72B (Team, 2025), which demonstrates strong mathematical reasoning capabilities and friendly deployment requirements. Specifically, we split each model response using a predefined list of discourse markers, including *Wait, But, Alternatively, Hmm, However, Let*, which prior work has shown to signal reasoning transitions or overthinking behaviors (Chen et al., 2024; Wang et al., 2025a; Sui et al., 2025). These markers naturally segment the reasoning trace into semantically coherent chunks. For each chunk, we prompt Qwen2.5-72B with the original problem, ground-truth answer, and the chunk itself (using the prompt in Fig. 5) to judge whether the chunk semantically contains the correct answer. We filter out responses that violate the expected output format and collect valid annotations until reaching 5,000 unique problems. To assess annotation quality, we manually verify 100 randomly sampled (chunk, judgment) pairs and find no clear misclassifications, indicating high reliability of the teacher model's labels. We then fine-tune a Qwen2.5-1.5B-Instruct model on this dataset for 2 epochs via supervised learning, constraining its output to `\boxed{yes|no}` for efficient online inference. The model is served via vLLM (Kwon et al., 2023) using the same prompt during training. As shown in Fig. 6c, the detector achieves high accuracy ($>99\%$) on the development set and stabilizes quickly during training. Given its strong agreement with human judgment and consistent performance, we treat its predictions as a reliable proxy for the ground-truth NRP position in our downstream training pipeline.

**Generalization Tests**   To test its generalization, we evaluated the NRP detector's predictions on model generations from two out-of-domain benchmarks: GPQA-Diamond (science) and Live-CodeBench (coding), under the DeepSeek-R1-Distill-1.5B model. Three expert annotators independently assessed the correctness of predictions from the NRP detector. The overall evaluation protocol is the same as illustrated above, where the NRP detector classifies whether a reasoning chunk contains the correct final answer and the human expert judges whether such classification is correct. The prediction is incorrect only if (1) a chunk containing a correct answer is classified as "False" and (2) a chunk without a correct answer is classified as "True". We compute the classification accuracy as the proportion of chunks correctly labeled across 100 correct responses from each dataset in Table 6. These results confirm that the math-trained NRP detector exhibits near-perfect reliability on science questions and very high ($>97\%$) accuracy on coding tasks. This high classification accuracy provides a solid foundation for the observed zero-shot transfer success of the full DECS framework.

## H   ADDITIONAL EXPERIMENTS

### H.1   DETERMINATION OF $\beta$ IN EQ. 11

We conduct a grid search on the AIME2024 dev set on both base models. We search the $\beta$ in the following range: $[0.0, 0.1, 0.2, 0.3, 0.5]$ and the results in Table 9 indicate that the value 0.2 achieves a great trade-off of efficiency gains and performance maintenance, where 0.0 represents that we only takes the decoupled-reward and follows Yu et al. (2025) to filter out extremely easy or hard prompts. Meanwhile, the optimal value of $\beta$ obtained in the 1.5B scale model transfers to the 7B model, which demonstrates that the value 0.2 is robust. We deem that as the policy's initial ratio of NRP $\mathcal{R}_0$ is approximately 0.5, a value of 0.2 guarantees that there will be at most $(100 - 50) \cdot 0.2 = 10$ percent of easy prompts among a batch. This quantity ensures that the condition in Theorem 1 is

---

[2]https://huggingface.co/datasets/open-r1/OpenR1-Math-220k

> **NRP Detector Prompt**
>
> ```
> Given a math problem and a segment of a long reasoning
> process to solve the problem, your task is to identify
> whether this segment has presented a correct final answer.
> If the segment contains information that can serve as the
> final answer to the problem and the answer is semantically
> correct when referring to the ground truth, simply explain
> the reason and output \boxed{yes}.  Otherwise, directly
> output \boxed{no}.
> **Problem**:
> {problem}
> **Reasoning segment**:
> {segment}
> **Ground Truth**:
> {answer}
> ```

Figure 5: Prompt for the training and inference with the NRP detector

Table 9: Grid search result on AIME2024 for different $\beta$ values.

| Model | $\beta$ | 0.00 | 0.10 | 0.20 | 0.30 | 0.50 | Base |
|-------|---------|------|------|------|------|------|------|
| 1.5B | Pass@1 | 32.98 | 31.77 | 31.25 | 29.23 | 25.42 | 27.99 |
|      | #Tokens | 7960 | 6876 | 5550 | 5497 | 5019 | 12202 |
| 7B   | Pass@1 | 57.29 | 54.16 | 51.33 | 50.00 | 43.96 | 50.65 |
|      | #Tokens | 8277 | 7525 | 5277 | 5114 | 4905 | 10508 |

hardly satisfied during the training progress. Although a more fine-grained search value like 0.25 may bring a better trade-off, we leave it for future research.

## H.2 TRAINING LOGS

In this section, we demonstrate the training curves of DECS on the 1.5B model and 7B model on Fig. 6 and Fig. 7, respectively. We select AIME2024 as a representative evaluation set, and plot the average reward and response length every 5 steps. Moreover, we also plot the average response length and the proportion of NRP (PNRP) during training to show that DECS achieves superior efficiency gains by reducing a large amount of non-NRP tokens in the thinking process.

## H.3 ABLATION WITH OTHER RL ALGORITHMS

Apart from GRPO, REINFORCE++ (Hu et al., 2025a) is also another strong algorithm for RLVR. Therefore, we also experiment DECS by taking REINFORCE++ (R++) to estimate the advantage value and use the same update formula as Eq. 3. Specifically, we use the same reward design as Eq. 9 but change the advantage estimator as below:

$$A_{i,j}^n = r_{i,j} - \text{mean}(r_{1,j}, \cdots, r_{G,j})$$

$$\hat{A}_{i,j}^n = \frac{A_{i,j}^n - \text{mean}(A)}{\text{std}(A)}$$

$$\text{mean}(A) = \frac{1}{\sum_{n=1}^{B} \sum_{i=1}^{G} |\boldsymbol{o}_i^n|} \sum_{n=1}^{B} \sum_{i=1}^{G} \sum_{j=1}^{|\boldsymbol{o}_i^n|} A_{i,j}^n$$

$$\text{std}(A) = \sqrt{\frac{\sum_{n=1}^{B} \sum_{i=1}^{G} \sum_{j=1}^{|\boldsymbol{o}_i^n|} (A_{i,j}^n - \text{mean}(A))^2}{(\sum_{n=1}^{B} \sum_{i=1}^{G} |\boldsymbol{o}_i^n|) - 1}}$$

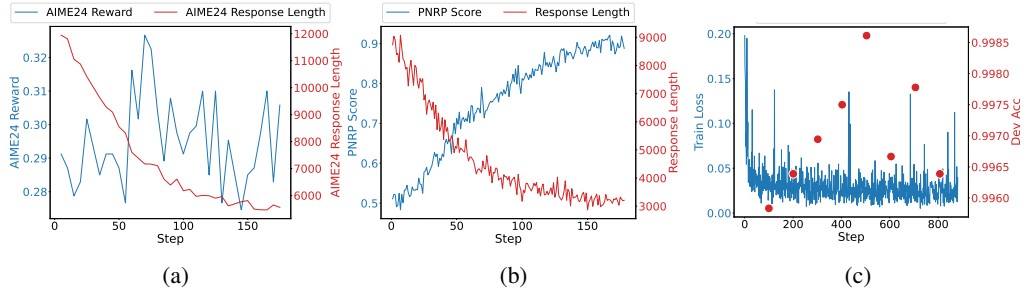

Figure 6: (a) AIME2024 reward and response length during evaluation for training DeepSeek-R1-Distill-1.5B base model with DECS and (b) Proportion of NRP (PNRP) and response length during training for training DeepSeek-R1-Distill-1.5B base model with DECS. (c) The training log and accuracy on the dev set of the trained NRP detector.

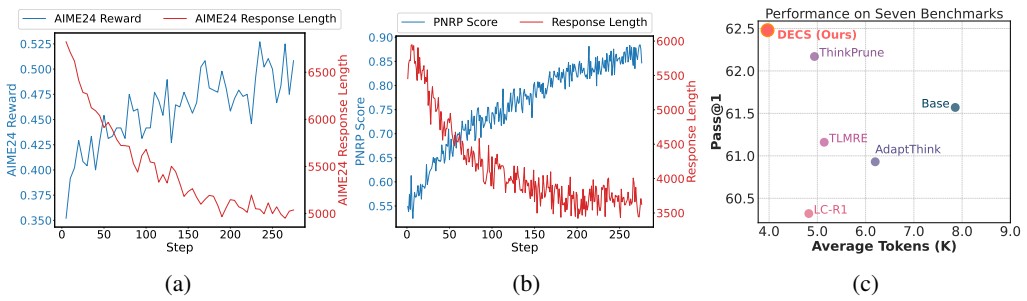

Figure 7: (a) AIME2024 reward and response length during evaluation for training DeepSeek-R1-Distill-7B base model with DECS; (b) Proportion of NRP (PNRP) and response length during training for training DeepSeek-R1-Distill-7B base model with DECS; (c) DECS improves pass@1 of base models while reducing $\sim 50\%$ tokens compared to the 7B base model across 7 benchmarks.

where $\hat{A}_{i,j}^n$ is the advantage for the $j$-th token of $i$-th rollout generated based on $n$-th prompt among a batch size of $B$. Results in Table 10 illustrate that there is no significant difference between GRPO and R++, which verifies the robustness of DECS.

## H.4 COMPARED TO MORE EFFICIENT REASONING BASELINES

Apart from the four baselines presented in Table 1, in this section, we compare more baselines that adopt the other approaches, different from length rewards, to improve the reasoning efficiency. We select S-GRPO (Dai et al., 2025), VSPO (Yue et al., 2025), MinD (Zeng et al., 2025b), LASER (Liu et al., 2025) and LAPO (Wu et al., 2025). We also include the over-sampling baseline that filters prompts whose rollouts are all correct or incorrect. Table 11 demonstrates that DECS outperforms other baselines on the seven benchmarks at both efficiency and efficacy, which further validates the effectiveness of DECS.

Table 10: Ablation on the other strong algorithm: REINFORCE++ with DECS on the DeepSeek-R1-Distill-1.5B base model. The REINFORCE++ variant achieves similar performance and efficiency improvements compared to using GRPO, validating the generality of DECS.

| Model | AIME2024 | | AIME2025 | | AMC23 | | MATH500 | | OlympiadB | | GPQA-D | | LCB | |
|---|---|---|---|---|---|---|---|---|---|---|---|---|---|---|
| | Acc | #Tok. | Acc | #Tok. | Acc | #Tok. | Acc | #Tok. | Acc | #Tok. | Acc | #Tok. | Acc | #Tok. |
| Base | 27.99 | 12202 | 22.94 | 12138 | 69.84 | 7875 | **84.55** | 4847 | 53.78 | 9217 | 32.86 | 8540 | 24.53 | 10560 |
| DECS w/ GRPO | **31.25** | **5550** | 23.78 | **4965** | 75.37 | 2988 | 84.40 | 1817 | 56.10 | 3396 | 35.92 | **3255** | 27.66 | **6026** |
| DECS w/ R++ | 30.75 | 5595 | **24.40** | 4978 | **75.68** | **2912** | 84.50 | **1770** | **56.23** | 3381 | **35.95** | 3333 | **27.85** | 6043 |

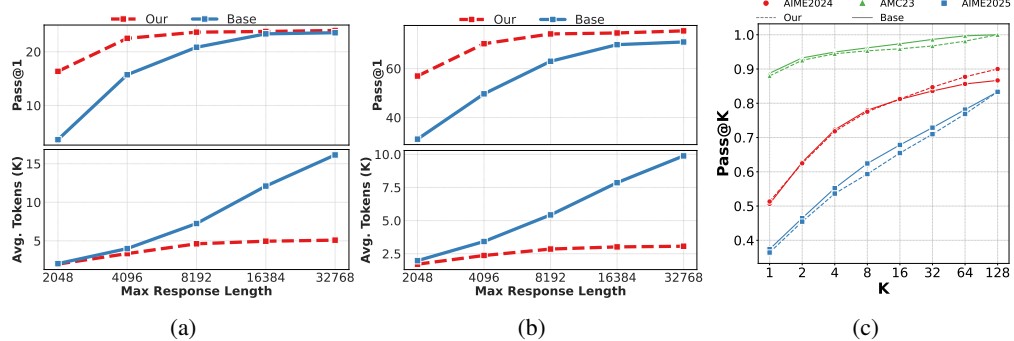

(a)                          (b)                          (c)

Figure 8: The Pass@1 score and average token counts on (a) AIME2025 and (b) AMC23 datasets under diverse token limits with the DeepSeek-R1-Dsitill-1.5B base policy; (c) Models applying DECS are on par with the base policy (DS-7B) in terms of Pass@K scores on three challenging benchmarks.

Table 11: Comparison with more methods targeted at efficient reasoning. DECS outperforms other baselines consistently across seven benchmarks.

| Model | AIME2024 | | AIME2025 | | AMC23 | | MATH500 | | OlympiadB | | GPQA-D | | LCB | |
|---|---|---|---|---|---|---|---|---|---|---|---|---|---|---|
| | Acc | #Tok. | Acc | #Tok. | Acc | #Tok. | Acc | #Tok. | Acc | #Tok. | Acc | #Tok. | Acc | #Tok. |
| *DS-1.5B* | | | | | | | | | | | | | | |
| Base | 27.99 | 12202 | 22.94 | 12138 | 69.84 | 7875 | **84.55** | 4847 | 53.78 | 9217 | 32.86 | 8540 | 24.53 | 10560 |
| MinD | 27.14 | 6172 | 21.46 | 6094 | 69.70 | **2883** | 82.80 | **1719** | 52.86 | 3573 | 31.30 | 4690 | 25.95 | 7217 |
| VSRM-R++ | 29.38 | 6954 | 22.58 | 6671 | 72.41 | 3633 | 84.10 | 2241 | 54.77 | 4388 | 33.44 | 4413 | 26.58 | 7377 |
| LAPO | 29.00 | 6936 | 22.20 | 6554 | 73.13 | 3770 | 84.30 | 2354 | 55.13 | 4530 | 34.34 | 4579 | 26.10 | 7033 |
| LASER-D | 28.83 | 5966 | 22.20 | 5584 | 73.55 | 3058 | 84.20 | 1872 | 55.28 | 3676 | 34.29 | 3863 | 26.42 | 6493 |
| DECS | **31.25** | 5550 | **23.78** | 4965 | **75.37** | 2988 | 84.40 | 1817 | **56.10** | 3396 | **35.92** | 3255 | **27.66** | 6026 |
| *DS-7B* | | | | | | | | | | | | | | |
| Base | 50.65 | 10508 | **36.67** | 11096 | 88.77 | 5764 | **93.25** | 3654 | 69.22 | 7507 | 46.46 | 7502 | 45.95 | 8966 |
| MinD | 49.57 | 9441 | 35.16 | 9997 | 87.22 | 4833 | 91.60 | 2859 | 68.17 | 6457 | 46.18 | 6528 | 45.63 | 8293 |
| VSRM-R++ | 47.68 | 6773 | 32.56 | 6953 | 84.66 | 3704 | 89.80 | 2044 | 66.13 | 5470 | 45.16 | 5764 | 44.89 | 7525 |
| S-GRPO | 50.93 | 7371 | 36.01 | 7908 | 88.20 | 3605 | 92.40 | 2252 | 69.74 | 4746 | 47.87 | 4938 | 47.34 | 7316 |
| LASER-D | 50.89 | 6423 | 35.61 | 6935 | 87.87 | 2949 | 92.20 | 1836 | 69.74 | 3914 | 48.18 | 4205 | 47.71 | 6789 |
| DECS | **51.33** | 5277 | 36.43 | 5516 | **89.04** | 2772 | 92.96 | 1728 | **70.28** | 3283 | **49.27** | 3276 | **48.05** | 5921 |

## I   CASE STUDY

We here present the comparison between DECS and three baselines, LC-R1, ThinkPrune and Adapt-Think, on MATH500, GPQA-Diamond, and LiveCodeBench, respectively. To conduct a compre-

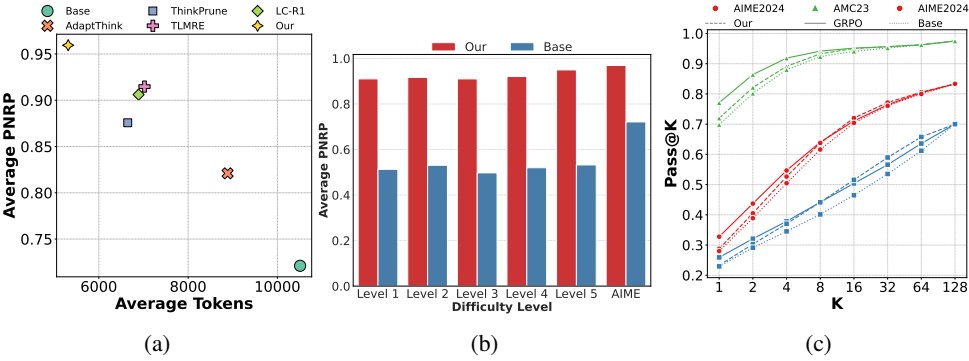

(a)                          (b)                          (c)

Figure 9: (a) The comparison between PNRP score and token consts in AIME2024 dataset for methods applied to the DS-7B model. (b) The PNRP scores for the six levels of difficulty on math problems for the DeepSeek-R1-Distill-7B base policy. (c) The Pass@K comparison between Base, Ours (DECS) and GRPO in DS-1.5B backbone.

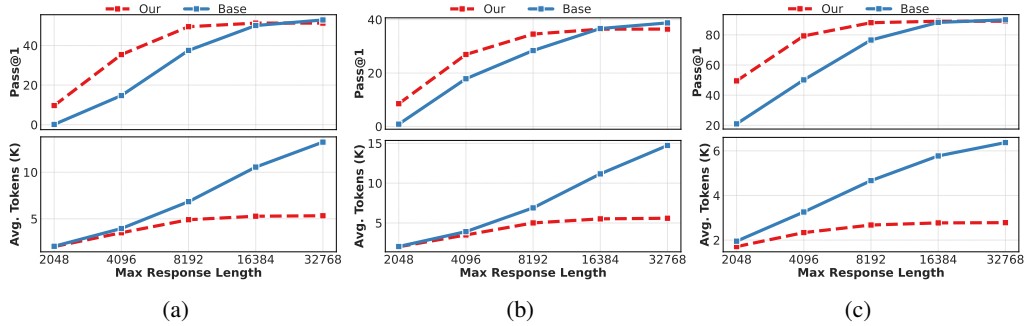

Figure 10: The Pass@1 score and average token counts on (a) AIME2024, (b) AIME2025 and (c) AMC23 datasets under diverse token limits with the DeepSeek-R1-Dsitill-7B base policy;

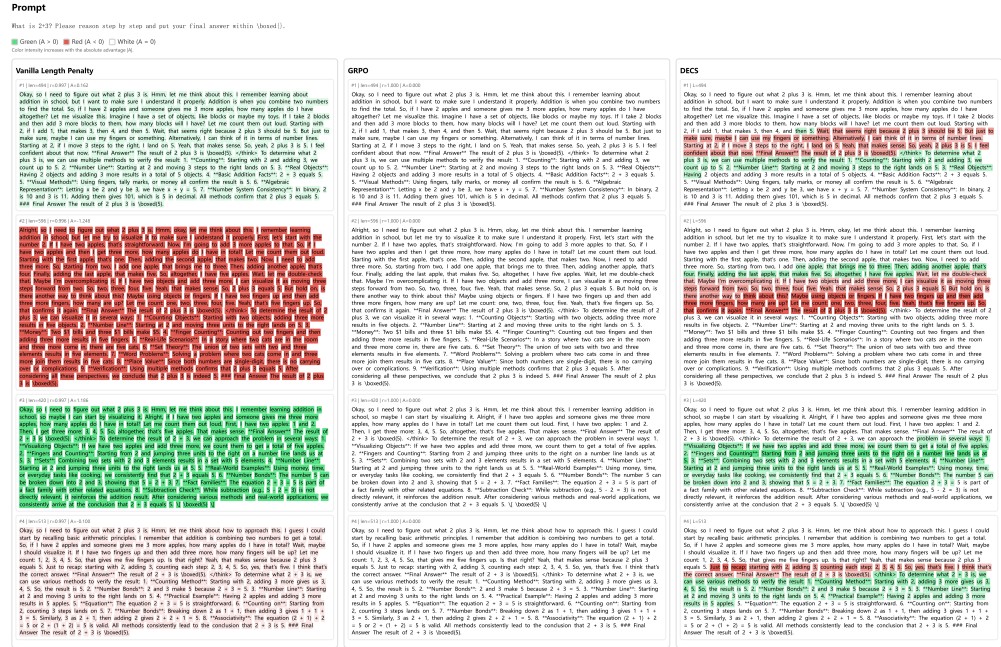

Figure 11: Illustrative example for advantage assignment under vanilla length penalty (left), GRPO (middle) and DECS (right) given a simple question "What is 2+3?". When applying vanilla length penalty, the algorithm would penalize the whole sequence for longer sequences (the second sequence), while rewarding redundant tokens for short sequences (the first response). However, DECS always penalizes unnecessary reasoning parts following the necessary reasoning prefix, no matter how short the whole sequence is (the first and last response). Additionally, the vanilla length penalty would penalize high entropy tokens in longer sequences, e.g., 2nd and 4th response, which deteriorates the model's exploration potentials throughout the training process.

hensive evaluation, we compare the outputs with LC-R1 taking DeepSeek-R1-Distill-1.5B as the base policy on MATH500, compare the outputs with ThinkPrune taking DeepSeek-R1-Distill-7B as the base policy on GPQA-Diamond, and comare the outputs with AdaptThink taking DeepSeek-R1-Distill-7B as the base policy on LiveCodeBench. The comparisons are illustrated in Fig. 13, Fig. 14 and Fig. 15, respectively.

## Token Entropy

**Temperature**: 0.6  **Cache Samples**: 6  **Analyzed tokens**: 494

Mean H (bits): **0.805**    Min: **0.000**    Max: **5.031**

**Prompt**

```
What is 2+3? Please reason step by step and put your final answer within \boxed{}.
```

**Highlighted By Entropy (full text)**

Okay, so I need to figure out what 2 plus 3 is. Hmm, let me think about this. I remember learning about addition in school, but I want to make sure I understand it properly. Addition is when you combine two numbers to find the total. So, if I have 2 apples and someone gives me 3 more apples, how many apples do I have altogether?

Let me visualize this. Imagine I have a set of objects, like blocks or maybe my toys. If I take 2 blocks and then add 3 more blocks to them, how many blocks will I have? Let me count them out loud. Starting with 2, if I add 1, that makes 3, then 4, and then 5. Wait, that seems right because 2 plus 3 should be 5. But just to make sure, maybe I can use my fingers or something.

Alternatively, I can think of it in terms of number lines. Starting at 2, if I move 3 steps to the right, I land on 5. Yeah, that makes sense.

So, yeah, 2 plus 3 is 5. I feel confident about that now.

**Final Answer**
The result of 2 plus 3 is \boxed{5}.
</think>

To determine what 2 plus 3 is, we can use multiple methods to verify the result:

1. **Counting**: Starting with 2 and adding 3, we count up to 5.
2. **Number Line**: Starting at 2 and moving 3 steps to the right lands on 5.
3. **Real Objects**: Having 2 objects and adding 3 more results in a total of 5 objects.
4. **Basic Addition Facts**: 2 + 3 equals 5.
5. **Visual Methods**: Using fingers, tally marks, or money all confirm the result is 5.
6. **Algebraic Representation**: Letting x be 2 and y be 3, we have x + y = 5.
7. **Number System Consistency**: In binary, 2 is 10 and 3 is 11. Adding them gives 101, which is 5 in decimal.

All methods confirm that 2 plus 3 equals 5.

### Final Answer
The result of 2 plus 3 is \boxed{5}.

Figure 12: Illustrative example showing high-entropy forking tokens. The distribution is similar to Wang et al. (2025b), where uncertainty-based tokens including "Wait", "but", and "maybe" have much larger entropy values than deterministic tokens like "5" which is the final answer.

**Question**

Define $p = \sum_{k=1}^{\infty} \frac{1}{k^2}$ and $q = \sum_{k=1}^{\infty} \frac{1}{k^3}$. Find a way to write $\sum_{j=1}^{\infty} \sum_{k=1}^{\infty} \frac{1}{(j+k)^3}$

**Ours Longest: 1724 Tokens**  **Ground Truth Answer: $p$-$q$**

**_NRP: 1191 Tokens_**

Okay, so I have this problem where I need to find an expression for the double sum $\sum_{j=1}^{\infty} \sum_{k=1}^{\infty} \frac{1}{(j+k)^3}$ in terms of $p$ and $q$, where $p = \sum_{k=1}^{\infty} \frac{1}{k^2}$ and $q = \sum_{k=1}^{\infty} \frac{1}{k^3}$. Hmm, okay, let's break this down step by step.\nFirst, I know that $p$ is the well-known Basel problem, which sums to $\frac{\pi^2}{6}$, but I don't know if that's relevant here. Maybe not directly, but it's good to remember. The other sum, $q$, is the Riemann zeta function at 3, so it's $\zeta(3)$, but again, not sure if that's useful yet.\nThe main goal is to express the double sum $\sum_{j=1}^{\infty} \sum_{k=1}^{\infty} \frac{1}{(j+k)^3}$ in terms of $p$ and $q$. I need to find a clever way to relate this double sum to the single sums $p$ and $q$.\n\nI remember that sometimes, for sums like this, changing variables or using symmetry can help. Let me try to visualize the double sum. It's over all positive integers $j$ and $k$, and the term is $\frac{1}{(j+k)^3}$. So, for each pair $(j, k)$, we're adding up the reciprocal of the cube of their sum.\n\nI wonder if I can express this double sum as a single sum by changing the order of summation or by some substitution. Maybe if I let $n = j + k$, then for each $n$, I can sum over all pairs $(j, k)$ such that $(j + k = n)$. That seems promising.\n\nSo, let me try that. Let me fix $n$ and consider all pairs $(j, k)$ where $(j + k = n)$. For each $n$, $j$ can range from 1 to $n - 1$, with $k = n - j$. Therefore, the double sum can be rewritten as:\n\n\[\sum_{n=2}^{\infty} \sum_{j=1}^{n-1} \frac{1}{n^3}\]\n\nBecause $(j + k = n)$ implies $(k = n - j)$, and since both $j$ and $k$ are at least 1, $n$ starts from 2 (since the smallest $j$ and $k$ can be is 1, so $1 + 1 = 2$).\n\nSo, simplifying the inner sum, it's just $\sum_{j=1}^{n-1} \frac{1}{n^3}$. Since $\frac{1}{n^3}$ is constant with respect to $j$, the inner sum is just $\frac{n-1}{n^3}$.\n\nTherefore, the entire double sum becomes:\n\n\[\sum_{n=2}^{\infty} \frac{n-1}{n^3}\]\n\nI can split this fraction:\n\n\[\sum_{n=2}^{\infty} \left( \frac{n}{n^3} - \frac{1}{n^3} \right) = \sum_{n=2}^{\infty} \left( \frac{1}{n^2} - \frac{1}{n^3} \right)\]\n\nNow, I can split this into two separate sums:\n\n\[\sum_{n=2}^{\infty} \frac{1}{n^2} - \sum_{n=2}^{\infty} \frac{1}{n^3}\]\n\nI know that the original definitions of $p$ and $q$ start from $n = 1$, so I can write these sums in terms of $p$ and $q$ by subtracting the $n=1$ terms.\n\nSo, $\sum_{n=2}^{\infty} \frac{1}{n^2} = p - \frac{1}{1^2} = p - 1$, and similarly, $\sum_{n=2}^{\infty} \frac{1}{n^3} = q - \frac{1}{1^3} = q - 1$.\n\nTherefore, substituting back into our expression:\n\n\[(p - 1) - (q - 1) = p - 1 - q + 1 = p - q\]\n\nSo, the double sum simplifies to $p - q$.\n\nI think that makes sense. So, the key steps were changing variables to $n = j + k$, then rewriting the double sum as a single sum over $n$, evaluating the inner sum, and then expressing the result in terms of $p$ and $q$. The important part was recognizing that the inner sum simplifies nicely and then using the definitions of $p$ and $q$ to relate it back.\nI don't see any mistakes in the steps, so I'm pretty confident the answer is $p - q$.\n\n**Final Answer**\nThe double sum can be written as $\boxed{p - q}$.\

**_Remaining: None_**

**NRP Ratio: 100%**

**LC-R1 Longest: 4287 Tokens**

**_NRP: 3283 Tokens_**

Okay, so I need to find a way to express the double sum $\sum_{j=1}^{\infty} \sum_{k=1}^{\infty} \frac{1}{(j+k)^3}$ in terms of $p$ and $q$, where $p = \sum_{k=1}^{\infty} \frac{1}{k^2}$ and $q = \sum_{k=1}^{\infty} \frac{1}{k^3}$. Hmm, let me think about how to approach this.\n\nFirst, I know that $p$ is the well-known Basel problem, which sums to $\frac{\pi^2}{6}$, but I don't think I need the exact value here. Similarly, $q$ is the sum of reciprocals of cubes, which converges to Apéry's constant, approximately 1.202..., but again, exact value isn't necessary here.\n\nThe double sum given is over all positive integers $j$ and $k$, and the term inside is $\frac{1}{(j+k)^3}$. So, it's a double series where each term depends on the sum of two variables. This seems a bit tricky because it's a two-dimensional sum, unlike single sums which are straightforward.\n\nMaybe I can change variables or find a way to express this double sum in terms of $p$ and $q$. Let me consider switching the order of summation or using some symmetry.\n\nAnother idea: perhaps express $1/(j + k)^3$ as an integral. Since $(1/n^3)$ can be written as an integral, maybe I can use that to interchange the sum and the integral.\n\nRecall that for positive integers $n$, we have:\n\n\[\frac{1}{n^3} = \frac{1}{2} \int_0^1 \int_0^1 (xy)^{n-1} dx\, dy\]\n\nWait, is that correct? Let me check. Actually, [omitted…] So, that gives us $p - q$.

**_Remaining: 492 Tokens_**

Hmm, that seems too straightforward.\n\nWait, but is that correct? [omitted…] \nWhich is equal to:\n\n\[\left( \sum_{m=1}^{\infty} \frac{1}{m^2} - 1 \right) - \left( \sum_{m=1}^{\infty} \frac{1}{m^3} - 1 \right) = (p - 1) - (q - 1) = p - q\n\]\n\nSo, is that correct? Hmm. Let me verify.[omitted…] which is indeed $p - q$.\n\nTherefore, the double sum is equal to $p - q$.\n\nSo, the answer is $\boxed{p - q}$.\n\n**Final Answer**\n\boxed{p - q}\n

**NRP Ratio: 87%**

Figure 13: Case study of the comparison of DECS and LC-R1 in MATH500.

**Question**

A spin-half particle is in a linear superposition $0.5|\uparrow\rangle + \sqrt{3}/2|\downarrow\rangle$ of its spin-up and spin-down states. If $|\uparrow\rangle$ and $|\downarrow\rangle$ are the eigenstates of $\sigma_z$, then what is the expectation value up to one decimal place, of the operator $10\sigma_z + 5\sigma_x$? Here, symbols have their usual meanings\n\nA) -1.4\nB) -0.7\nC) 1.65\nD) 0.85

Ground Truth Answer: *B)*

**Ours Longest: 2766 Tokens**

*NRP: 2294 Tokens*

Okay, so I have this quantum mechanics problem here, and I need to find the expectation value of the operator $10\sigma z + 5\sigma x$ for a given spin-½ particle state. Hmm, let me try to break this down step by step.\nFirst, the state of the particle is given as a linear combination of the spin-up and spin-down states: $0.5|\uparrow\rangle + (\sqrt{3}/2)|\downarrow\rangle$. I remember that $|\uparrow\rangle$ and $|\downarrow\rangle$ are eigenstates of the Pauli matrix $\sigma z$. So, $\sigma z$ acting on $|\uparrow\rangle$ gives +1 times $|\uparrow\rangle$, and $\sigma z$ acting on $|\downarrow\rangle$ gives -1 times $|\downarrow\rangle$, right?\n\nNow, the operator we're dealing with is $10\sigma z + 5\sigma x$. I need to find the expectation value of this operator, which is $\langle\psi| (10\sigma z + 5\sigma x) |\psi\rangle$. Since expectation value is linear, I can separate this into $10\langle\psi|\sigma z|\psi\rangle + 5\langle\psi|\sigma x|\psi\rangle$. That should make things easier.\n\nLet me handle each term separately. Starting with the first term, $10\langle\psi|\sigma z|\psi\rangle$. I know that $\sigma z$ is diagonal in the $\sigma z$ basis, which is the same as the $|\uparrow\rangle$ and $|\downarrow\rangle$ basis. So, the expectation value of $\sigma z$ is just the weighted average of the eigenvalues, with the weights given by the square of the amplitudes in the state $|\psi\rangle$.\n\nThe state $|\psi\rangle$ is $0.5|\uparrow\rangle + (\sqrt{3}/2)|\downarrow\rangle$. So, the coefficient for $|\uparrow\rangle$ is 0.5, and for $|\downarrow\rangle$ is $\sqrt{3}/2$. The square of these gives the probabilities. Therefore, the expectation value of $\sigma z$ is $(0.5)^2*(+1) + (\sqrt{3}/2)^2*(-1)$. Calculating that:\n\n(0.25)(1) + (3/4)(-1) = 0.25 - 0.75 = -0.5.\n\nSo, 10 times that would be 10*(-0.5) = -5.\n\nOkay, that's the first part. Now, the second term is $5\langle\psi|\sigma x|\psi\rangle$. Hmm, $\sigma x$ is the Pauli X matrix, which is not diagonal in the $\sigma z$ basis. I remember that $\sigma x$ has eigenstates $|+x\rangle$ and $|-x\rangle$, which are superpositions of $|\uparrow\rangle$ and $|\downarrow\rangle$. Maybe I need to express $|\psi\rangle$ in the $\sigma x$ basis to compute this expectation value, or perhaps there's another way.\n\nAlternatively, I recall that $\sigma x$ can be written in matrix form as [[0,1],[1,0]]. So, if I can write $|\psi\rangle$ as a column vector, then $\sigma x$ acting on $|\psi\rangle$ can be computed, and then the inner product $\langle\psi|\sigma x|\psi\rangle$ can be found.\n\nLet's represent $|\psi\rangle$ as a vector. Since $|\uparrow\rangle$ is [1; 0] and $|\downarrow\rangle$ is [0; 1], then $|\psi\rangle$ is $0.5*[1; 0] + (\sqrt{3})/2*[0; 1] = [0.5; \sqrt{3}/2]$. So, the vector is $[0.5, \sqrt{3}/2]^T$.\n\nNow, $\sigma x$ is [[0,1],[1,0]]. So, $\sigma x|\psi\rangle$ is $[0.5*1 + (\sqrt{3}/2)*0, 0.5*0 + (\sqrt{3}/2)*1]^T$, which is $[0.5, \sqrt{3}/2]^T$. Wait, that's the same as $|\psi\rangle$? Hmm, so $\sigma x|\psi\rangle = |\psi\rangle$? [omitted] Since a and b are real in this case (the coefficients are real), this becomes 2ab.\n\nSo, for our state, a = 0.5, b = $\sqrt{3}/2$. Therefore, $\langle\sigma x\rangle = 2*(0.5)*(\sqrt{3}/2) = (\sqrt{3})/2 \approx 0.866$.\n\nTherefore, $5*\langle\sigma x\rangle = 5*(\sqrt{3}/2) \approx 5*0.866 \approx 4.33$.\n\nSo, adding the two terms together: $-5 + 4.33 \approx -0.67$. So, approximately -0.7 when rounded to one decimal place.\n\nAh, okay, so that makes sense now. My initial mistake was probably in the way I computed $\sigma x|\psi\rangle$ and then the inner product. But using the formula for expectation value in terms of coefficients, it's much simpler. So, the expectation value of $\sigma x$ is 2ab, which is $2*(0.5)*(\sqrt{3}/2) = \sqrt{3}/2 \approx 0.866$. Therefore, 5 times that is approximately 4.33, and 10 times the expectation value of $\sigma z$ is -5.\n\nSo, adding them gives approximately -0.67, which rounds to -0.7. So, the answer should be option **B) -0.7**.

*Remaining: None*

NRP Ratio: 100%

**ThinkPrune Longest: 8098 Tokens**

*NRP: 1535 Tokens*

Okay, so I have this quantum mechanics problem here about a spin-half particle. Hmm, let's see. The particle is in a linear superposition given by $0.5|\uparrow\rangle + (\sqrt{3})/2|\downarrow\rangle$. And I need to find the expectation value of the operator $10\sigma\_z + 5\sigma\_x$. The answer choices are A) -1.4, B) -0.7, C) 1.65, D) 0.85.\n\nAlright, first, let me recall what the expectation value of an operator is. For a quantum state $|\psi\rangle$, the expectation value $\langle O\rangle$ is given by $\langle\psi| O |\psi\rangle$. So, I need to compute $\langle\psi| (10\sigma\_z + 5\sigma\_x) |\psi\rangle$.\n\nSince expectation value is linear, I can separate this into $10\langle\psi|\sigma\_z|\psi\rangle + 5\langle\psi|\sigma\_x|\psi\rangle$. So, I need to find the expectation values of $\sigma\_z$ and $\sigma\_x$ separately and then combine them.\n\nLet me write down the given state $|\psi\rangle$. It is $0.5|\uparrow\rangle + (\sqrt{3})/2|\downarrow\rangle$. Let me note that $|\uparrow\rangle$ and $|\downarrow\rangle$ are the eigenstates of $\sigma\_z$, which is the Pauli z-matrix. So, $\sigma\_z|\uparrow\rangle = +|\uparrow\rangle$ and $\sigma\_z|\downarrow\rangle = -|\downarrow\rangle$.\n\nFirst, compute $\langle\sigma\_z\rangle$. That is, $\langle\psi|\sigma\_z|\psi\rangle$.\n\nSince $\sigma\_z$ is diagonal in the $\sigma\_z$ basis, which is the same as the basis in which $|\uparrow\rangle$ and $|\downarrow\rangle$ are defined, this should be straightforward. [omitted…]Looking at the options, B is -0.7, which would be the answer.\n\nBut wait, let me verify the calculations again because sometimes I might have made a mistake.\n\nFirst, $\langle\sigma\_z\rangle$: probabilities 0.25 and 0.75, so 0.25 - 0.75 = -0.5. That seems right.\n\nThen, for $\langle\sigma\_x\rangle$: the expectation value.\n\nAnother way to compute $\langle\sigma\_x\rangle$ is to note that $\sigma\_x = |\rightarrow\rangle\langle\rightarrow| - |\leftarrow\rangle\langle\leftarrow|$, but maybe that's more complicated.\n\nAlternatively, for a general state $|\psi\rangle = a|\uparrow\rangle + b|\downarrow\rangle$, $\langle\sigma\_x\rangle$ is $|a|^2 - |b|^2$? Wait, no, that can't be.\n\nWait, $\sigma\_x$ is [[0,1],[1,0]], so when you compute $\langle\psi|\sigma\_x|\psi\rangle$, it's $|a|^2*\langle\downarrow|\sigma\_x|\uparrow\rangle + |b|^2*\langle\uparrow|\sigma\_x|\downarrow\rangle$ + cross terms.\n\nWait, let's compute it again step by step.\n\nIn $|\psi\rangle = [0.5; (\sqrt{3})/2]$. So, in matrix form, it's a column vector [0.5; $(\sqrt{3})/2$].\n\n$\sigma\_x |\psi\rangle = [ (\sqrt{3})/2 ; 0.5 ]$\n\nThen, $\langle\psi| \sigma\_x |\psi\rangle$ is [0.5, $(\sqrt{3})/2$] multiplied by [ $(\sqrt{3})/2$ ; 0.5 ]\n\nSo, $0.5*(\sqrt{3})/2 + (\sqrt{3})/2 *0.5 = (\sqrt{3})/4 + (\sqrt{3})/4 = \sqrt{3} / 2 \approx 0.866$\n\nSo, that seems correct.\n\nTherefore, $5\langle\sigma\_x\rangle = 5*(\sqrt{3})/2 \approx 4.3301$, and $10\langle\sigma\_z\rangle = -5$, so total expectation value is $-5 + 4.3301 \approx -0.6699$, which is approximately -0.67, so -0.7 when rounded to one decimal.\n\nTherefore, the answer should be **B) -0.7**.

*Remaining: 5990 Tokens*

\n\nWait, but let me think again. Is there another way to compute this? Maybe using the properties of spin states. [omitted...] So, in conclusion, $\langle\sigma\_x\rangle$ is approximately 0.866, which is $\sqrt{3} / 2$.\n\nTherefore, $5\langle\sigma\_x\rangle$ is $5*(\sqrt{3} / 2) \approx 5*0.866 \approx 4.33$\n\nThen, $10\langle\sigma\_z\rangle = -5$, as computed earlier.\n\nTherefore, the total expectation value is $-5 + 4.33 \approx -0.67$, which is approximately -0.7 when rounded to one decimal place.\n\nTherefore, the answer is **B) -0.7**.

NRP Ratio: 20%

Figure 14: Case study of the comparison of DECS and ThinkPrune in GPQA-Diamond.

---

**Question**

Slavic is preparing a present for a friend's birthday. He has an array $a$ of $n$ digits and the present will be the product of all these digits. Because Slavic is a good kid who wants to make the biggest product possible, he wants to add 1 to exactly one of his digits. \n\nWhat is the maximum product Slavic can make?\n\nInput\n\nThe first line contains a single integer $t(1 \leq t \leq 10^4)$— the number of test cases.\n\nThe first line of each test case contains a single integer $n(1 \leq n \leq 9)$— the number of digits.\n\nThe second line of each test case contains $n$ space-separated integers $a_i$ $(0 \leq a_i \leq 9)$— the digits in the array.\n\nOutput\n\nFor each test case, output a single integer— the maximum product Slavic can make, by adding 1 to exactly one of his digits.Sample Input 1:\n4\n\n4\n\n2 2 1 2\n\n3\n\n0 1 2\n\n5\n\n4 3 2 3 4\n\n9\n\n9 9 9 9 9 9 9 9 9\n\n\n\nSample Output 1:\n\n16\n2\n432\n430467210\n"

---

**Ours Longest: 2932 Tokens**

**NRP: 2471 Tokens**

Okay, I need to solve this problem where Slavic can add 1 to exactly one digit in an array of digits to maximize the product. Hmm, let's think about how to approach this.\n\nSo, the problem is: given an array of digits (each between 0 and 9), I can choose one digit to add 1 to. Then, compute the product of all the digits. I need to find the maximum possible product after this operation.\n\nFirst, I need to [omitted...]Sample Input 1:\n\n4\n\n4\n\n2 2 1 2\n\nSo, the array is [2,2,1,2]. For each position, adding 1 and compute product.\n\nLet's see:\n\nOption 1: add to first 2 → [3,2,1,2]. Product is 3*2*1*2=12.\n\nOption 2: add to second 2 → same as option 1, product 12.\n\nOption 3: add to 1 → [2,2,2,2]. Product is 2^4=16.\n\nOption4: add to last 2 → [2,2,1,3]. Product is 2*2*1*3=12.\n\nSo maximum is 16. Which matches the sample output.\n\nAnother sample:\n\n3\n\n0 1 2\n\nPossible options:\n\nAdd 1 to 0 → [1,1,2] → product 1*1*2=2.\n\nAdd 1 to 1 → [0,2,2] → 0*2*2=0.\n\nAdd 1 to 2 → [0,1,3] → 0*1*3=0.\n\nSo maximum is 2. Which matches sample.\n\nAnother sample: 5 digits 4,3,2,3,4.\n\nWhat's the best option?\n\nAdding 1 to the 2: makes it 3, so the array becomes [4,3,3,3,4]. Product is 4*3*3*3*4 = 4*3^3 *4 = 4*27 *4 = 432.\n\nWhich matches sample.\n\nAnother sample: 9 9s. Adding 1 to one of them gives 10, others 9. So product is 10 * 9^8.\n\nCompute 9^8: 9*9=81, 81*9=729, 729*9=6561, 6561*9=59049, 59049*9=531441, 531441*9=4782969, 4782969*9=43046721. Multiply by 10: 430467210. Which matches the sample.\n\nSo the code should handle all cases correctly.\n\nNow, in Python, the code can be written as follows.\n\nRead t:\n\nt = int(input())\n\nfor _ in range(t):\n    n = int(input())\n    a = list(map(int, input().split()))\n\n    max_prod = 0\n\n    for i in range(n):\n        new_a = a.copy()\n        new_a[i] += 1\n        product = 1\n        for num in new_a:\n            product *= num\n        if product > max_prod:\n            max_prod = product\n\nprint(max_prod)\n\nYes, that's the code.

**Remaining: None**

**NRP Ratio: 100%**

---

**AdaptThink Longest: 3797 Tokens**

**NRP: 1349 Tokens**

Okay, so I've got this problem to solve. Let's read it carefully.\n\nThe problem is about Slavic who wants to maximize the product of digits in an array after adding 1 to exactly one of the digits. So, for each test case, I need to find which digit to increment by 1 such that the product becomes as large as possible.\n\nHmm, let's think about the approach. For each digit in the array, I can consider increasing it by 1 and then calculate the product. Then, I pick the maximum among all these possibilities.\n\nWait, but the array could have a zero. Oh right, because if a digit is 9, adding 1 makes it 10. But if a digit is 0, adding 1 makes it 1, which might be better in some cases.\n\nLet me outline the steps:\n\n1. For each test case:\n   a. Read the array of digits.\n   b. For each digit in the array, create a copy where that digit is increased by 1. But wait, if the digit is 9, adding 1 would make it 10. But since we're treating each digit as a single digit, maybe I should represent them as integers. So, for each element a_i, I'll compute a_i + 1, but only if a_i is 9, it becomes 10. [omitted...]\n\nSo the code structure would be something like:\n\nfor _ in range(t):\n    n = int(input())\n    a = list(map(int, input().split()))    max_prod = 0\n    for i in range(n):\n        new_a = a.copy()\n        new_a[i] += 1\n        current_prod = 1\n        for num in new_a:\n            current_prod *= num\n        if current_prod > max_prod:\n            max_prod = current_prod\n    print(max_prod)

**Remaining: 2026 Tokens**

Wait, but wait. What if the original array has a zero, and adding 1 to it? [omitted]\n\nNow, for the code:\n\nRead t.\n\nFor each test case:\n\n    Read n.\n\n    Read the list as integers.\n\n    Initialize max_prod to zero.\n\n    For each i in 0..n-1:\n\n        new_a = a.copy()\n\n        new_a[i] +=1\n\n        compute product of new_a.\n\n        if product is larger than current max_prod, update.\n\n        print max_prod.\n\nBut wait, in the case where all digits are zero except one. [omitted]\n\nSo the code:\n\nt = int(input())\n\nfor _ in range(t):\n    n = int(input())\n    a = list(map(int, input().split()))\n    max_product = 0\n    for i in range(n):\n        new_a = a.copy()\n        new_a[i] += 1\n        product = 1\n        for num in new_a:\n            product *= num\n        if product > max_product:\n            max_product = product\n    print(max_product)\n\nWait, but wait. What if the array is [0, 0, 0], the original product is 0. Adding 1 [omitted]\n\nSo the code should be correct.\n\nThus, the code is as written above.\n

**NRP Ratio: 40%**

---

Figure 15: Case study of the comparison of DECS and AdaptThink in LiveCodeBench.

