# OpenReview forum: "Overthinking Reduction with Decoupled Rewards and Curriculum Data Scheduling"
_ICLR.cc/2026/Conference — ICLR 2026 Oral_

### Official Review · Reviewer_tgZx · 2025-10-23

**Soundness:** 4
**Presentation:** 2
**Contribution:** 3
**Rating:** 10
**Confidence:** 2

**Summary:**

Handling RLVR model’s overthinking behavior by separating tokens according to their role and penalizing redundant tokens, and scheduling batch with curriculum.
–
Prior approaches incorporating length penalties (e.g., Eq. 4) broadcast trajectory-level rewards to all tokens via GRPO's advantage estimation (Eq. 2). When all responses in a group are correct but vary in length, shorter sequences receive positive advantages while longer ones receive negative advantages uniformly across every token. Consequently, high-entropy tokens like "Wait" or "However"—which initiate productive exploration—are suppressed simply because they appear in longer trajectories, even when they are part of the necessary reasoning process (Lemma 2, Theorem 1). DECS addresses this by decoupling rewards at the token level: it assigns maximum reward (r₊) to all NRP tokens including high-entropy exploratory tokens, while inversely penalizing only the redundant tokens generated after the NRP (Eq. 9). This surgical approach maintains exploratory capacity (preserving Pass@K performance, Fig. 3c) while achieving superior compression (>50% token reduction, Table 1).

**Strengths:**

- The idea is backed by theoretical claims, and they are logically correct. The two flaws revealed by Lemma 2 and Theorem 2 (suppression of high-entropy tokens + failure to penalize redundant tokens) are addressed by DECS through curriculum scheduling (Theorem 1) and decoupled rewards.
- In addition, the paper make each claim irrefutable through quantitative verification at every level - from theory to core analysis (Fig 1, 3), then validate broadly with 7 benchmarks x 2 models x 5+ baselines, with robustness checks (Tables 5-7)
- The paper organized their contents very nicely, well distributed content between main text and appendix.
- Good generalization to out-of-domain tasks (LiveCodeBench, GPQA-D)

**Weaknesses:**

Limited accessibility for non-expert readers: While the paper presents rigorous theoretical analysis and strong empirical results, the presentation could be more accessible. Figures 1-2, which are central to understanding the core contributions, are dense with notation that is not immediately intuitive without careful reading of Section 3. For a paper introducing novel concepts like "Necessary Reasoning Prefix" and "decoupled token-level rewards," more pedagogical support would strengthen impact.

Suggestions for improvement:
- Add a concrete, annotated example in Appendix showing token-by-token reward assignment for a simple problem (e.g., "What is 2+3?"). Compare how (a) base GRPO, (b) length penalty, and (c) DECS assign advantages to specific tokens including high-entropy ones.
- Figure 2, while comprehensive, relies heavily on abstract labels ("NRP", "Redundancy", "Correct Thinking Process") that may not be
immediately intuitive to readers unfamiliar with the concepts. Consider replacing these placeholders with concrete text snippets from an actual model response (e.g., showing "Wait, let me verify... 2+3=5" as NRP vs "Let me triple-check... yes, 5" as redundancy) within the same figure space. This would make the core contribution—that DECS protects exploratory tokens while pruning redundancy—immediately graspable without requiring back-reference to Section 3.
- Include an intuitive explanation of why high-entropy tokens matter, possibly with a probability distribution visualization showing how entropy differs between "5" (low H) vs "Wait" (high H).

**Questions:**

* Q1: The paper presents strong technical contributions, but accessibility could be improved with concrete worked examples (e.g., token-by-token walkthrough of the "2+3" problem showing advantages under baseline vs DECS). Given the paper is already dense, where do you envision such pedagogical content fitting best?
* Q2: Table 1 shows 7B has less overthinking than 1.5B initially. Does this trend continue? Would larger models need DECS?  (Scalability to 30B+ models unverified)

---

> ### Author Response · Authors · 2025-11-17
> **Response to Reviewer tgZx**
>
> Thank you sincerely for your exceptionally positive evaluation and the outstanding score of 10! We are pleased to hear that you find our proposed DeCS  solid, novel, and robust with theoretical support! Below, we follow your suggestions to revise our paper and address your concerns:
>
> > W1.1, Q1 Add a concrete, annotated example in Appendix showing token-by-token reward assignment for a simple problem (e.g., "What is 2+3?"). Compare how (a) base GRPO, (b) length penalty, and (c) DECS assign advantages to specific tokens including high-entropy ones.
>
> **Our response**: Thanks for your suggestions. In Appendix Figure 11 of the revised manuscript, we add an example to illustrate how DECS assigns token-level reward for "What is 2+3?". The example shows that DECS always penalizes unnecessary reasoning parts following the necessary reasoning prefix, no matter how short the whole sequence is.
>
> > W1.2  Replacing technical placeholders with concrete text snippets from an actual model response
>
> **Our response**: Thanks for your suggestions. We follow your suggestions to replace the placeholder with actual reasoning segments for the problem "What is 2+3?" in the revised manuscript.
>
> > W1.3 Include an intuitive explanation of why high-entropy tokens matter, possibly with a probability distribution visualization showing how entropy differs between "5" (low H) vs "Wait" (high H).
>
> **Our response**: Thanks for your suggestions! We add a probability distribution visualization in Figure 12 in the appendix of the revised manuscript.
>
> > Q2: Table 1 shows 7B has less overthinking than 1.5B initially. Does this trend continue? Would larger models need DECS? (Scalability to 30B+ models unverified)
>
> **Our response**: Thank you for this insightful question. We appreciate the opportunity to clarify an important point: the severity of overthinking should not be measured solely by total reasoning length, but rather by the proportion of necessary reasoning prefix (PNRP) — a metric we introduce in §6 (RQ1) to quantify how much of the generated trace is truly essential for correctness.
>
> While it's true that the 7B model outputs fewer absolute tokens than the 1.5B model (Table 1), both of them contain substantial redundancy. To investigate whether this trend continues on larger scales, we evaluate two 30B+ models — DeepSeek-R1-Distill-32B and Qwen3-32B — on AIME2024, AIME2025, and AMC23 using the same inference settings as in §5.1. The results are summarized below:
>
> | Model       | AIME2024 pass@1 | AIME2024 #Toks | AIME2024 PNRP | AIME2025 pass@1 | AIME2025 #Toks | AIME2025 PNRP | AMC23 pass@1 | AMC23 #Toks | AMC23 PNRP |
> |-------------|------------------|---------------|--------------|----------------|---------------|--------------|--------------|------------|-----------|
> | DS-32B      | 64.79            | 9690.66       | 0.72         | 48.33          | 10721.18      | 0.72         | 92.66        | 5433.37    | 0.61      |
> | Qwen3-32B   | 70.42            | 10746.01      | 0.61         | 56.04          | 12105.69      | 0.59         | 92.50        | 6646.34    | 0.52      |
>
> These results show that even at the 30B+ scale, at least 30–40% of reasoning tokens are redundant (PNRP < 0.7). This indicates that overthinking remains a pervasive issue across model sizes.
>
> Notably, DECS achieves PNRP > 0.95 while reducing tokens by over 50% (Fig. 3b, Fig. 9a), suggesting significant room for improvement even in larger models. Although we currently lack the computational resources to train DECS on 30B+ models, the consistency of PNRP degradation across scales strongly motivates its application to larger systems.
>
> We are actively working on efficient implementations and collaborative efforts to scale DECS, and will release training guidance to support community replication.

---

### Official Review · Reviewer_YPs8 · 2025-10-28

**Soundness:** 2
**Presentation:** 3
**Contribution:** 1
**Rating:** 2
**Confidence:** 3

**Summary:**

The paper addresses the "overthinking" problem in large reasoning models trained with reinforcement learning, where models generate excessively long reasoning paths without performance benefits. The authors propose DECS (Decoupled rewards and Curriculum data Scheduling) to reduce the reasoning length while not harming the performance.

**Strengths:**

- The studied problem is important;
- The writing is easy to follow;

**Weaknesses:**

- The proposed reward and curriculum designs are quite artificial. In particular, the decoupled reward depends on pre-training an effective NRP detection model, which introduces practical challenges given that the definition of NRP is ambiguous, let alone the difficulty of reliably detecting it with another model.
- The method is not straightforward to implement, as it requires pre-training a separate detection model whose accuracy cannot be guaranteed or easily verified.
- The curriculum design component lacks novelty and, as such, does not constitute a substantial contribution to the paper.

**Questions:**

See the weaknesses.

---

> ### Author Response · Authors · 2025-11-17
> **Response to Reviewer YPs8 (Part1)**
>
> Thank you for your review and important questions! We are glad to hear that you found our proposed theoretical analysis useful and DeCS solid, innovative and robust with sufficient experiments. Below we address the concerns and questions raised in the review.
>
> > Weakness 1: The proposed reward and curriculum designs are quite artificial. In particular, the decoupled reward depends on pre-training an effective NRP detection model, which introduces practical challenges given that the definition of NRP is ambiguous, let alone the difficulty of reliably detecting it with another model.
>
> > Weakness 2: The method is not straightforward to implement, as it requires pre-training a separate detection model whose accuracy cannot be guaranteed or easily verified.
>
> **Our response**: We thank the reviewer for raising these combined concerns, which address the core practicality of the Necessary Reasoning Prefix (NRP) module. We respectfully assert that the NRP concept is **rigorously defined, highly reliable, and eminently practical**.
> 1. **Clarity and Validity of the NRP Definition**
>
> The Necessary Reasoning Prefix (NRP) is defined in our work as the **minimal prefix of a reasoning trajectory that logically justifies the correct answer for the first time (Definition 1)**. Any tokens generated after this point are, by definition, redundant.
> - **Empirical Evidence**: As illustrated in **Figure 11**, the LC-R1 baseline generated the correct answer at token 3283 but continued with 492 tokens of redundant "self-doubt." The first 3283 tokens form the NRP—a complete, valid derivation—and all subsequent tokens can be safely truncated.
> - **Theoretical Validation**: The principled nature of the NRP concept has been acknowledged by multiple reviewers (e.g., "Solid Theoretical Foundation," "The paper provides a clear theoretical analysis..."). The NRP is a non-ambiguous, principled target for efficiency optimization.
> 2. **Reliability and Generalization of the NRP Detector**
>
> The practicality concern centers on the accuracy of the detection model. We provide extensive evidence for its reliability and generalizability:
> - **High Accuracy**: The lightweight NRP detector (Qwen2.5-1.5B) is trained on high-quality data generated by a strong teacher (Qwen2.5-72B) and verified with human annotation on 100 random samples. As demonstrated in **Figure 6(c)**, the detector achieves **over 99% accuracy** on the development set, confirming its consistency and learnability.
> - **Zero-Shot Generalization (Practicality Proof)**: The true test of the detector's utility is its ability to generalize, making it highly practical for diverse applications. Despite being trained only on math data, when the full DECS framework is applied to out-of-domain tasks, it outperforms previous state-of-the-art:
>   - On the **GPQA-Diamond (Science)** benchmark, DECS reduces reasoning tokens by **61.88%** (with DS-1.5B) and **56.33%** (with DS-7B) while improving performance.
>   - On the **LiveCodeBench (Coding)** benchmark, DECS reduces tokens by **42.94%** (with DS-1.5B) and  **33.96%** (DS-7B) while improving performance.
>
> This cross-domain efficacy is compelling evidence that the detector captures a **robust and transferable notion of sufficient reasoning progress**, not just a math-specific pattern. The required implementation step (pre-training the detector) is straightforward and yields a highly effective, verified module.

---

> > ### Author Response · Authors · 2025-11-17
> > **Response to Reviewer YPs8 (Part2)**
> >
> > > W3: The curriculum design component lacks novelty and, as such, does not constitute a substantial contribution to the paper.
> >
> > **Our response**: We thank the reviewer for the feedback and appreciate the opportunity to clarify the **novelty and contribution** of our curriculum prompt scheduling component.
> > Our design is not a generic application of curriculum learning, but a direct, theory-informed solution to a specific and previously unaddressed problem in efficient reasoning: the **undue suppression of high-entropy tokens** essential for valid exploration under sequence-level length penalties (Section 3.2, Theorem 1).
> > - **Addressing a Theoretical Flaw**: Prior work treats length regularization as a static reward term, which inadvertently creates a gradient bias against necessary exploratory behavior. Our key insight is that this issue must be mitigated by **dynamically controlling the data distribution** during training.
> > - **Adaptive, Novel Mechanism**: Our curriculum scheduler is innovative because it is the **first method** to use **online, fine-grained reasoning structure analysis** (via NRP estimation) to adaptively schedule training prompts in critic-free RL. It creates a self-regulating loop where the scheduler uses the model’s learned redundancy level (NRP ratio) to automatically adjust the proportion of easy prompts in each batch (Eq. 11, Section 4.3).
> > - **Substantial Contribution**: The efficacy of this mechanism is confirmed by our ablation study (**Figure 3a**), which shows that removing the scheduler results in a significant **1.47-point drop in Pass@1 score**. This confirms the scheduler is a critical component that enables DECS to achieve over 50% token reduction **without sacrificing performance**.
> >
> > In summary, the scheduler is an integral, novel part of our framework, specifically designed to overcome a fundamental limitation of existing RL approaches, thus representing a key contribution.
> >
> > > Response to Low Contribution Score (Contribution: 1; Rating: 2, reject)
> >
> > We appreciate the reviewer’s feedback and would like to respectfully highlight the key contributions of our work, which represent a meaningful advance in the design of efficient reasoning models:
> > 1. **Theoretical Novelty**: To the best of our knowledge, this is the first work to formally identify and analyze two fundamental flaws in current length-rewarded RL methods (as detailed in Appendix A):
> >    - Sequence-level length penalties inadvertently suppress high-entropy tokens essential for valid reasoning (Theorem 1).
> >    - Misaligned incentives lead to spurious rewards for partial redundancy, hindering convergence (Theorem 2).
> > 2. **Methodological Innovation**: Based on these findings, we propose DECS, a novel framework featuring:
> >    - A **decoupled token-level reward** that surgically penalizes only redundant tokens after the NRP, avoiding the erroneous over-penalization of exploratory behavior.
> >    - A **novel, adaptive curriculum prompt scheduling strategy** (Eq. 11) that dynamically adjusts data composition based on on-the-fly NRP detection, balancing efficiency and exploration.
> > 3. **Empirical Rigor and Generalization**: We conduct extensive evaluations across 7 benchmarks and 2 model scales (1.5B & 7B), showing that DECS consistently reduces reasoning tokens by **over 50%** while maintaining or improving performance (Pass@1, Pass@K) compared to all baselines. Furthermore, the newly added generalization experiments in the revised manuscript on the Qwen3 model family (Section 5.4) and the human studies on the NRP detector's cross-domain accuracy (Appendix G.6) solidify the broad applicability of our approach.
> >
> > We believe that the combination of our novel theoretical analysis, the innovative two-part methodological solution (decoupled reward and adaptive curriculum), and the consistent, significant empirical gains across diverse domains collectively constitutes a strong and substantial contribution to the field of efficient LLM reasoning.

---

> ### Author Response · Authors · 2025-11-24
> **Sincerely looking forward to more discussion with you**
>
> Dear Reviewer,
>
> Thank you again for your thoughtful feedback and for taking the time to engage with our work. We truly appreciate your attention to the practical aspects of our method. In our rebuttal, we have carefully addressed your concerns, providing additional empirical evidence, theoretical clarification, and cross-domain validation to strengthen the presentation.
>
> We realized that some aspects of our approach—particularly the NRP detection module and the adaptive curriculum design—may not have been presented with sufficient clarity in the original submission, partly because we introduced several new mathematical notations and abbreviations in a compact form. We have now streamlined the notation and exposition, which we hope makes it easier to see the method’s novelty and practicality. In response to your comments as well as those from the other reviewers, we have substantially revised the paper to better highlight the theoretical insights and methodological innovations behind DECS.
>
> In particular, we have expanded our discussion of the rigor of the NRP definition, added human-verified evaluation of detector accuracy to address the practicality and verifiability of the decoupled reward, and included new generalization results on Qwen3, which further demonstrate the robustness and broader applicability of our framework. We also clarify how the adaptive curriculum design differs from standard curriculum or reward-shaping schemes and provide additional analyses to isolate its contribution.
>
> We hope that these clarifications and revisions help to more clearly convey the contributions of our work, and we would be very grateful if you could consider our updated manuscript and detailed rebuttal when convenient.
>
> Thank you once again for your valuable feedback, which has helped us improve the paper substantially.
>
> Best regards,
> On behalf of all authors of submission 3706

---

### Official Review · Reviewer_fsTY · 2025-10-29

**Soundness:** 4
**Presentation:** 4
**Contribution:** 4
**Rating:** 6
**Confidence:** 4

**Summary:**

This paper introduces DECS, a novel framework designed to mitigate "overthinking", the generation of excessively long reasoning paths, in large reasoning models (LRMs) trained with reinforcement learning. The authors first provide a theoretical analysis identifying two key flaws in current length-penalty methods: the erroneous penalization of essential exploratory tokens and the inadvertent rewarding of redundant tokens. To address these, DECS incorporates three main components: (1) a lightweight judge model to detect the Necessary Reasoning Prefix (NRP), (2) a decoupled token-level reward that specifically penalizes tokens generated after the NRP, and (3) a scheduling strategy to balance efficiency and performance during training. The authors conduct extensive experiments, demonstrating that DECS can reduce the number of reasoning tokens while maintaining or even improving performance.

**Strengths:**

1. The paper provides a clear theoretical analysis of why existing length-reward mechanisms are suboptimal.
2. The DECS offers a carefully designed approach grounded in theoretical reasoning.
3. The authors evaluate their method across diverse benchmarks, use two different model scales, and compare against multiple relevant and strong baselines.
4. DECS achieves a reduction in token count while simultaneously improving performance.

**Weaknesses:**

1. The framework's reliance on a separate, fine-tuned NRP detector adds complexity to the training pipeline. A deeper analysis of its robustness would strengthen the paper.
2. The decoupled reward is used to differentiate between "Leading Redundant tokens" and other redundant tokens, but this does not appear to be reflected in Eq.9.

**Questions:**

1. I am intrigued by the parallels between the "erroneous penalization" issue and concepts like relative overgeneralization in MARL; a brief comment on this could enrich the paper's context within the broader RL literature.
2. Could the authors elaborate on the reward design shown in Fig.2 and Eq.9?

---

> ### Author Response · Authors · 2025-11-17
> **Response to Reviewer fsTY (Part1)**
>
> Thank you for your review and important questions! We are glad to hear that you found our proposed theoretical analysis useful and DeCS solid, innovative and robust with sufficient experiments. Below we address the concerns and questions raised in the review.
>
> > [W1] Deeper analysis of NRP detector robustness
>
> **Our response**: We sincerely thank the reviewer for this insightful comment, which allows us to highlight a key strength of our framework: the remarkable zero-shot generalization capability of DECS across diverse reasoning domains.
> While the NRP detector is trained on mathematical reasoning traces from OpenR1-Math-220K, its core function—identifying the necessary reasoning prefix (NRP) that first contains the final answer—is a fundamental aspect of reasoning itself, not just mathematical reasoning. The true test of its utility lies in whether DECS, built upon this math-specialized detector, can achieve superior efficiency and performance on out-of-domain tasks.
>
> Critically, our main results already demonstrate this powerful transferability. As shown in Table 1, when applied to the DeepSeek-R1-Distill-7B model:
> - On the LiveCodeBench (coding) benchmark, DECS achieves a higher Pass@1 accuracy than all baselines while using around 10% fewer tokens.
> - On the GPQA-Diamond (science) benchmark, DECS outperforms all baselines, with over 27% fewer tokens compared to the most efficient baseline.
>
> These results are compelling evidence that the NRP concept learned in the math domain effectively transfers to coding and scientific reasoning. DECS successfully identifies and eliminates redundancy in these very different contexts, achieving state-of-the-art efficiency without sacrificing (and often improving) performance. This cross-domain efficacy strongly suggests that the underlying signal captured by the NRP detector is robust and generalizable.
>
> To directly address your specific request for an analysis of the NRP detector's classification performance, we have conducted a new human evaluation study on non-mathematical domains. We evaluated the detector's predictions on generations from GPQA-Diamond (science) and LiveCodeBench (coding) using the DeepSeek-R1-Distill-1.5B model. Three expert annotators independently judged the correctness of the detector's binary classification (whether a chunk contains the correct final answer). The overall accuracy, calculated as the proportion of correctly labeled chunks across 100 correct responses per dataset, is summarized below:
>
> | Dataset       | Human 1 | Human 2 | Human 3 |
> |---------------|---------|---------|---------|
> | GPQA-D        | 99.18   | 98.96   | 99.23   |
> | LiveCodeBench | 97.88   | 97.63   | 97.49   |
>
> These results confirm that the math-trained NRP detector exhibits near-perfect reliability on science questions and very high (>97%) accuracy on coding tasks. This high classification accuracy provides a solid foundation for the observed zero-shot transfer success of the full DECS framework.
>
> In summary, both the downstream task performance (Table 1) and the direct human evaluation of the detector converge to the same conclusion: the NRP detector, despite being math-specialized, learns a highly generalizable notion of reasoning progress, enabling DECS to be a broadly applicable solution for overthinking reduction.

---

> > ### Author Response · Authors · 2025-11-17
> > **Response to Reviewer fsTY (Part2)**
> >
> > [W2, Q2] The decoupled reward is used to differentiate between "Leading Redundant tokens" and other redundant tokens, but this does not appear to be reflected in Eq.9.
> >
> > **Our response**: We thank the reviewer for this insightful comment and for allowing us to clarify the subtle mechanism by which our reward system penalizes **all** redundant tokens, including the "leading redundant token" (the first token after the Necessary Reasoning Prefix, NRP), even though Eq. 9 does not explicitly differentiate them.
> > Our decoupled reward design, combined with the **Group-Relative Advantage computation** in DECS, ensures that **all leading redundant tokens consistently receive negative advantages** and are thus penalized during training. This property arises from two key aspects of our design:
> >
> > 1. **Reward Structure**: For any correct rollout, tokens within the NRP receive the maximum fixed reward $r^+=1.1$. Redundant tokens receive a lower, length-dependent discounted reward $r^0 + (r^+ - r^0)/L_i$, which is in the range $[1.0, 1.00625]$ (since $L_i \ge 16$ and the group size $G=16$).
> > 2. **Relative Advantage Mechanism**: A token receives a positive advantage only if its reward, $R_{i,j}$, exceeds the group average reward at that position, $\bar{R}^j$.
> >
> > We analyze the two possible scenarios for the leading redundant token in a group:
> > ### **Scenario 1: No Sequence Has a Longer NRP**
> >
> > This is the case where the sequence containing the leading redundant token has the **longest NRP** within the group (meaning all other sequences in the group have an equally long or shorter NRP).
> > - In this scenario, the reward of the leading redundant token, $R_{i,j}$, which is $r^0 + (r^+ - r^0)/L_{i}$, is **generally below the group average** $\bar{R}^j$.
> > - We empirically verified that this token consistently receives a negative advantage. We evaluated rollouts from the DS-1.5B/DS-7B models at various training steps and counted instances where the leading redundant token from the sequence with the longest NRP received an above-average reward.
> > | Step   | 1     | 60    | 120   | 180   | 240   |
> > |--------|-------|-------|-------|-------|-------|
> > | DS-1.5B| 0     | 0     | 0     | 0     | N/A   |
> > | DS-7B  | 0     | 0     | 0     | 0     | 0     |
> >
> > Across all observed batches and positions, **no instance** was found where a leading redundant token received a positive advantage using the Eq. 9 reward. This confirms that the decoupled reward structure effectively induces a negative advantage for this token.
> > ### **Scenario 2: At Least One Sequence Has a Longer NRP**
> >
> > If there exists at least one sequence with a longer or equally long NRP, then at least one token at the same position $t$ among the group receives the maximum reward, $r^+=1.1$.
> > - Since the maximum possible reward for any redundant token is $<1.00625$, and the group average reward $\bar{R}^j>(1*15+1.1)/16$ is necessarily $> 1.00625$ (due to the contribution of the full-NRP token), the redundant token’s reward is **strictly below the group average**.
> > - Therefore, the token **must receive a negative advantage** in this case.
> > Together, both the theoretical analysis (Scenario 2) and the empirical validation (Scenario 1) confirm that our decoupled reward design effectively prevents the erroneous rewarding of any leading redundant token. We have included this detailed explanation in **Appendix C** in the revised manuscript to be explicitly linked to Eq. 9.

---

> > > ### Author Response · Authors · 2025-11-17
> > > **Response to Reviewer fsTY (Part3)**
> > >
> > > > [Q1]: the parallels between the "erroneous penalization" issue and concepts like relative overgeneralization in MARL
> > >
> > > **Our response**: We thank the reviewer for this insightful comment and appreciate the observation regarding the connection between the "erroneous penalization" issue and concepts like **relative overgeneralization** in multi-agent reinforcement learning (MARL).
> > > While our work focuses on single-agent efficient reasoning in LLMs, we agree that the core problem, the **misalignment between global reward signals and local token-level updates**, resonates with broader challenges in the RL literature.
> > > - **The Parallel**: Relative overgeneralization in MARL occurs when agents learn suboptimal joint behaviors due to misleading credit assignment from shared rewards. Analogously, in our single-agent sequence setting, the coarse-grained, sequence-level length penalty acts as a blunt signal. This signal fails to accurately attribute cost to specific redundant tokens, leading to the **erroneous suppression of useful, high-entropy tokens**. This is essentially a form of **token-level overgeneralization**.
> > > - **The Solution**: This parallel highlights a fundamental challenge in policy gradient methods: the need for fine-grained, temporally precise feedback to avoid spurious credit assignment. Our proposed **decoupled reward mechanism** (Eq. 9) can be viewed as an instance of **structured credit assignment**. By explicitly disentangling necessary and redundant reasoning steps, we ensure that only truly redundant tokens are penalized, effectively mitigating this token-level overgeneralization.
> > >
> > > We thank the reviewer for prompting us to reflect on this deeper connection, and we have included a brief discussion in Appendix D in the revised version to situate our findings within this broader RL context.

---

> > > > ### Comment · Reviewer_fsTY · 2025-11-21
> > > >
> > > > Thanks to the authors for the detailed response and additional experimental results. The response has addressed my questions and concerns. Hence, I would raise my ratings accordingly.

---

> > > > > ### Author Response · Authors · 2025-11-21
> > > > >
> > > > > Thank you so much for your thoughtful feedback during the rebuttal. Your questions were really helpful in pushing us to clarify and strengthen the paper. We’re glad the revisions and new experiments helped address your concerns, and we truly appreciate your kind words and the time you put into reviewing our work!

---

### Official Review · Reviewer_XB3N · 2025-11-01

**Soundness:** 4
**Presentation:** 3
**Contribution:** 4
**Rating:** 8
**Confidence:** 4

**Summary:**

This paper targets the “overthinking” phenomenon on LRMs. Based on the fundamental misalignment between trajectory-level rewards and token-level optimization. The authors propose a DeCS framework based on theoretical discoveries that are erroneous penalization of essential exploratory tokens and  inadvertently rewarding partial redundancy. The DeCS framework includes (1) decoupled token-level reward mechanism and (2) curriculum batch scheduling strategy. Through comprehensive experiments the authors demonstrate effectiveness on their proposed framework

**Strengths:**

**\[S1\] Solid Theoretical Foundation**
The proposed framework is developed through strong theoretical foundation that enhance its novelty and provide convincing justification for its effectiveness. Also the paper's clear definition of concepts like NRP will likely provide significant value to the field by establishing precise terminology

**\[S2\] Comprehensive Experimentation and Analysis**
The paper presents a satisfying experimental structure organized around research questions, including thorough comparisons with baseline methods and detailed ablation studies. This approach effectively addresses potential questions about the proposed framework while convincingly demonstrating its effectiveness.

**Weaknesses:**

**\[W1\] Readability Issues**
While the logical flow of the paper is solid, it was difficult to understand due to the extensive use of abbreviations and mathematical notations. Additionally, the token length comparison in Table 1 would have been more effectively conveyed through visualization.

**\[W2\] Limited Model Generalization**
The experiments are confined exclusively to the DeepSeek model family. The paper would be strengthened by extending experimentation to model families with different exploration capabilities to demonstrate broader generalization of the proposed framework.

**\[W3\] Domain Dependency of the NRP Classification Model.**
While the NRP classifier trained on OpenR1-Math-220K demonstrates strong reliability through human validation within the mathematical domain, the paper lacks analysis of its performance in non-mathematical domains (e.g., coding, science, etc.). It will be great for the authors to additionally verify whether the NRP classification remains robust and consistent across these domains.

**Questions:**

**\[Q1\]** In the RQ2 experiment, I understand that the exploration potential maintains similar with the base model. I'm curious about how exploration compares between DeCS  and GRPO-trained models. What differences or similarities exist in the exploration behavior between DeCS and models trained with GRPO?

**\[Q2\]** Table 1 indicates that the proposed DeCS framework demonstrates lower average performance metrics compared to GRPO-trained models. Furthermore, the manuscript predominantly focuses on comparisons with baseline methods and analyses of individual DeCS components, while direct comparative analysis with GRPO appears somewhat limited. Given these observations, are there any theoretical reasons why DeCS shows a lower performance compared to GRPO?

---

> ### Author Response · Authors · 2025-11-17
> **Response to Reviewer XB3N (part1)**
>
> Thank you very much for your detailed reviews, comments, and suggestions. We are glad to hear that you find DeCS a solid method with a theoretical foundation and extensive experiments. Below, we address the concerns and questions raised in the review.
>
> > [W1] Readability Issues
>
> **Our Response**: We sincerely thank the reviewer for this valuable feedback on improving the paper's readability. We fully agree that clarity is paramount, and we have taken comprehensive steps in the revised manuscript to address these concerns:
>
> 1. **Abbreviations**: We have systematically reviewed the manuscript and ensured that every abbreviation (e.g., NRP, PNRP, LRM, DECS, GRPO) is explicitly defined upon its first appearance in each major section. We have also corrected the instances of previously undefined abbreviations in Figure 1 and the main text.
> 2. **Mathematical Notations**: To significantly enhance clarity, we have introduced a dedicated "Notation" table in the Appendix (Table 4). This table provides a clear, concise summary of all key mathematical symbols used. Furthermore, we now explicitly refer to this newly added table in Section 3 to provide clearer explanations for the notations upon their introduction.
> 3. **Visualization for Token Length Comparison**: We fully agree that the token length comparison is best conveyed visually. We have implemented the following changes to feature visual comparisons of token efficiency:
> - **Token Efficiency Comparison (DS-1.5B): Figure 1 (right)** has directly compared the **pass@1 score against the average token cost**, clearly illustrating DECS's superior efficiency on the DS-1.5B model.
> - **Extended Token Efficiency Comparison (DS-7B)**: Following this suggestion, we have added **Figure 7 (c)** in the revised manuscript, which compares the pass@1 score against average token cost on the DS-7B model. This visualization **conclusively shows that DECS achieves the highest pass@1 score with the fewest tokens.**
> These revisions significantly improve the manuscript's accessibility, and we believe the changes make our contributions and findings easier to follow.
>
> > [W2] Limited Model Generalization
>
> **Our response**: We appreciate the reviewer's suggestion to validate the generalization of our framework across different model families. While we initially focused on the DeepSeek family for consistent comparison with existing baselines, we fully agree on the importance of demonstrating broader applicability. As Table 1 has demonstrated that DeCS outperforms previous methods by a large margin in both model scales, we  conduct additional experiments on Qwen3-4B to show the effectiveness of DeCS on a stronger model:
> | Model       | AIME2024        |                 | AIME2025        |                 | AMC23           |                 | MATH500         |                 | OlympiadB       |                 | GPQA-D          |                 | LCB             |                 | Average         |                 |    |
> |-------------|-----------------|-----------------|-----------------|-----------------|-----------------|-----------------|-----------------|-----------------|-----------------|-----------------|-----------------|-----------------|-----------------|-----------------|-----------------|-----------------|-------|
> |             | Acc             | #Tok.           | Acc             | #Tok.           | Acc             | #Tok.           | Acc             | #Tok.           | Acc             | #Tok.           | Acc             | #Tok.           | Acc             | #Tok.           | Acc             | #Tok.           | AES    |
> | Qwen3-4B    | 64.82           | 11611           | 56.30           | 12870           | 91.60           | 7478            | 93.74           | 4839            | 71.07           | 9144            | 39.11           | 8072            | 62.17           | 9713            | 68.40           | 9104            | 0     |
> | w/ DeCS     | **65.38**       | **5431**        | **56.96**       | **5758**        | **93.59**       | **2864**        | **93.78**       | **1648**        | **74.09**       | **3646**        | **41.00**       | **3260**        | **63.27**       | **6196**        | **69.72**       | **4115**        | **0.61**  |
>
> The results demonstrate that DeCS improved the base model by 1.4 pass@1 on average, consuming only 54.8% tokens. This improvement is robust across 5 in-domain datasets and 2 domain out-of-domain datasets, which verifies that DeCS is a general method for different model backbones.
> We also include this experiment in Section 5.4 in the revised manuscript.

---

> ### Author Response · Authors · 2025-11-17
> **Response to Reviewer XB3N (part2)**
>
> > [W3] Domain Dependency of the NRP Classification Model.
>
> **Our response**: We sincerely thank the reviewer for this insightful comment, which allows us to highlight a key strength of our framework: the remarkable zero-shot generalization capability of DECS across diverse reasoning domains.
>
> While the NRP detector is trained on mathematical reasoning traces from OpenR1-Math-220K, its core function—identifying the necessary reasoning prefix (NRP) that first contains the final answer—is a fundamental aspect of reasoning itself, not just mathematical reasoning. The true test of its utility lies in whether DECS, built upon this math-specialized detector, can achieve superior efficiency and performance on out-of-domain tasks.
>
> Critically, our main results already demonstrate this powerful transferability. As shown in Table 1, when applied to the DeepSeek-R1-Distill-7B model:
> - On the LiveCodeBench (coding) benchmark, DECS achieves a higher Pass@1 accuracy than all baselines while using around 10% fewer tokens.
> - On the GPQA-Diamond (science) benchmark, DECS outperforms all baselines, with over 27% fewer tokens compared to the most efficient baseline.
>
> These results are compelling evidence that the NRP concept learned in the math domain effectively transfers to coding and scientific reasoning. DECS successfully identifies and eliminates redundancy in these very different contexts, achieving state-of-the-art efficiency without sacrificing (and often improving) performance. This cross-domain efficacy strongly suggests that the underlying signal captured by the NRP detector is robust and generalizable.
>
> To directly address your specific request for an analysis of the NRP detector's classification performance, we have conducted a new human evaluation study on non-mathematical domains. We evaluated the detector's predictions on generations from GPQA-Diamond (science) and LiveCodeBench (coding) using the DeepSeek-R1-Distill-1.5B model. Three expert annotators independently judged the correctness of the detector's binary classification (whether a chunk contains the correct final answer). The overall accuracy, calculated as the proportion of correctly labeled chunks across 100 correct responses per dataset, is summarized below:
>
> | Dataset       | Human 1 | Human 2 | Human 3 |
> |---------------|---------|---------|---------|
> | GPQA-D        | 99.18   | 98.96   | 99.23   |
> | LiveCodeBench | 97.88   | 97.63   | 97.49   |
>
> These results confirm that the math-trained NRP detector exhibits near-perfect reliability on science questions and achieves very high accuracy (>97%) on coding tasks. This high classification accuracy provides a solid foundation for the observed zero-shot transfer success of the full DECS framework.
>
> In summary, both the downstream task performance (Table 1) and the direct human evaluation of the detector converge to the same conclusion: the NRP detector, despite being math-specialized, learns a highly generalizable notion of reasoning progress, enabling DECS to be a broadly applicable, domain-flexible solution for reducing overthinking.
>
> > [Q1]: differences or similarities exist in the exploration behavior between DeCS and models trained with GRP
>
> **Our response**: We sincerely thank the reviewer for this insightful question. We agree that comparing the exploration behavior of a model trained for efficiency (DECS) against one trained purely for performance (GRPO) is highly meaningful.
>
> As shown in Figure 9(c) of the revised manuscript, we directly compare the pass@k (k=2,4,8,16,32,64,128) curves of the base model, the GRPO-trained model, and our DECS-trained model on multiple benchmarks. The results reveal a key finding: both the GRPO-trained model and our DECS-trained model exhibit nearly identical pass@k score to the base model when K is sufficiently large.
>
> **Similarity**: Both methods preserve the ability to find correct solutions through multiple stochastic rollouts, since neither of them penalizes correct high-entropy tokens, which is significant for exploration potentials.
>
> **Difference**: DECS, a method optimized for efficiency, could achieve similar exploration abilities to GRPO, which is an algorithm for maximizing correctness rewards. This demonstrates the superiority of the dynamic scheduling mechanism for maintaining sufficient exploration potentials.

---

> > ### Author Response · Authors · 2025-11-17
> > **Response to Reviewer XB3N (part3)**
> >
> > > [Q2]:  theoretical reasons why DeCS shows a lower performance compared to GRPO
> >
> > **Our response**: We thank the reviewer for this insightful question, which addresses the nuanced trade-off between maximizing raw performance and optimizing for efficiency. We respectfully clarify that the slight performance gap in Pass@1 is a **direct theoretical consequence** of the distinct optimization objectives of DECS versus GRPO, from the following two aspects:
> >
> > ### **Optimization Objectives and Performance Metrics**
> >
> > The fundamental distinction lies in the models' optimization goals:
> > - **GRPO** focuses **solely on maximizing correctness rewards**, prioritizing the highest possible $\text{Pass@1}$ score without explicit control over computational efficiency (token usage).
> > - **DECS** employs a **decoupled reward mechanism (Eq. 9)** and curriculum scheduling to **actively reduce redundancy** and minimize token usage, aiming for high performance under a constrained computation budget.
> > When evaluated on the comprehensive metric that measures both efficiency and performance, the **Average Efficiency Score (AES)**, DECS demonstrates clear superiority. For instance, DECS outperforms GRPO by **0.21** on the DS-1.5B model and by **0.31** on the DS-7B model in terms of $\text{AES}$. This confirms that DECS is the **most efficient overall framework** compared to GRPO.
> >
> > ### **Token Usage and Test-Time Scaling Law**
> >
> > The marginal Pass@1 advantage of GRPO can be partly explained by the well-known **test-time scaling law** (as noted in [1]), where increased reasoning length often correlates with higher accuracy. Crucially, **GRPO uses approximately 50% to 100% more tokens** on average than DECS. This dramatically increased computational budget provides GRPO with more opportunities for complex search, contributing to its marginally better Pass@1 score.
> > Therefore, the slightly lower Pass@1 score for DECS is not a weakness, but a testament to its primary design goal: **achieving highly competitive performance with dramatically reduced computational cost**. DECS successfully strikes the best balance between efficiency and performance among all methods tested.
> >
> > ## Reference
> >
> > [1] Snell C V, Lee J, Xu K, et al. Scaling LLM test-time compute optimally can be more effective than scaling parameters for reasoning[C]//The Thirteenth International Conference on Learning Representations. 2025.

---

> > > ### Comment · Reviewer_XB3N · 2025-11-19
> > >
> > > I greatly appreciate the authors' effort in providing such thorough explanations and conducting additional experiments to address the points I raised. The revisions to the manuscript are also well done.
> > >
> > > After carefully reviewing the detailed responses and updated manuscript, most of my concerns have been addressed.
> > >
> > > I still think the paper could be challenging for readers who are less familiar with this research area. However, for those in the field, this is a strong and valuable contribution that provides meaningful insights and I believe this paper deserves a spotlight or oral presentation, and I will update my overall score to 10 accordingly.
> > >
> > > Finally, full credit to the authors for conducting this excellent work!

---

> > > > ### Author Response · Authors · 2025-11-19
> > > >
> > > > Thank you so much for your thoughtful comments and the updated score. We're glad the revisions and additional experiments addressed your concerns. We truly appreciate your recognition of our work and the positive feedback. We'll keep your suggestion about accessibility for a broader audience in mind for future iterations. Thank you again for your time and support!

---

### Meta-Review · Area_Chair_XkWX · 2026-01-07

**Summary:**

This paper introduces a framework built on two technical innovations (decoupled token-level rewards and curriculum data scheduling) to address the overthinking phenomenon in large reasoning models. Reviewers generally highlight the solid theoretical foundation, methodological innovation, and thorough empirical analysis. Reviewer XB3N, tgZx, and fsTY strongly supported the paper after authors rebuttal addressed concerns regarding the robustness of the NRP detector, scalability, and clarity. Reviewer YPs8 raised concerns about the clarity of definitions, the practicality, and the novelty of the approach. Although the discussion ended unexpectedly early and further feedback was not possible, after carefully reading the reviews and discussion threads, I find the authors provided reasonable clarifications. Overall, given the consensus of most reviewers on the sound theoretical foundation and comprehensive experimental validation, I recommend acceptance.

---

Note: The automated analysis flagged the following reference as possibly incorrect:
- David Ruppert. The elements of statistical learning: data mining, inference, and prediction, 2004.

After checking, this appears to be a citation error. Please ensure the author names, title, and publication details are accurate in the final version.

**Reviewer Concerns:**

Reviewer YPs8 raised concerns about the clarity of definitions, practicality, and novelty of the approach. The author rebuttal explicitly addressed the clarity and validity of the NRP definition and its practical applicability. Regarding novelty, since no concrete prior work was identified that clearly diminishes the contribution, I view this as a more subjective assessment that, overall, does not outweigh the positive aspects of the paper.

**Reviewer Scores:**

Reviewer XB3N: 8 → 10. The reviewer stated: "I will update my overall score to 10 accordingly".\
Reviewer c: 10 → 10.\
Reviewer fsTY: 6 → 8. The reviewer stated: "I would raise my ratings accordingly".\
Reviewer YPs8: 2 → 4 or a positive score (AC's best guess).

---

### Decision · Program_Chairs · 2026-01-26

Accept (Oral)